# On Socially Fair Low-Rank Approximation and Column Subset Selection

**Zhao Song**
The Simons Institute for the Theory of Computing
UC Berkeley
magic.linuxkde@gmail.com

**Ali Vakilian**
Toyota Technological Institute at Chicago
vakilian@ttic.edu

**David P. Woodruff**
Department of Computer Science
Carnegie Mellon University
dwoodruf@andrew.cmu.edu

**Samson Zhou**
Department of Computer Science
Texas A&M University
samsonzhou@gmail.com

## Abstract

Low-rank approximation and column subset selection are two fundamental and related problems that are applied across a wealth of machine learning applications. In this paper, we study the question of socially fair low-rank approximation and socially fair column subset selection, where the goal is to minimize the loss over all sub-populations of the data. We show that surprisingly, even constant-factor approximation to fair low-rank approximation requires exponential time under certain standard complexity hypotheses. On the positive side, we give an algorithm for fair low-rank approximation that, for a constant number of groups and constant-factor accuracy, runs in $2^{\text{poly}(k)}$ time rather than the naïve $n^{\text{poly}(k)}$, which is a substantial improvement when the dataset has a large number $n$ of observations. We then show that there exist bicriteria approximation algorithms for fair low-rank approximation and fair column subset selection that run in polynomial time.

## 1 Introduction

Machine learning algorithms are increasingly used in technologies and decision-making processes that affect our daily lives, from high volume interactions such as online advertising, e-mail filtering, smart devices, or large language models, to more critical processes such as autonomous vehicles, healthcare diagnostics, credit scoring, and sentencing recommendations in courts of law [Cho17, KLL+18, BHJ+21]. Machine learning algorithms frequently require statistical analysis, utilizing fundamental problems from numerical linear algebra, especially low-rank approximation and column subset selection.

In the classical low-rank approximation problem, the input is a data matrix $\mathbf{A} \in \mathbb{R}^{n \times d}$ and an integral rank parameter $k > 0$, and the goal is to find the best rank $k$ approximation to $\mathbf{A}$, i.e., to find a set of $k$ vectors in $\mathbb{R}^d$ that span a matrix $\mathbf{B}$, which minimizes $\mathcal{L}(\mathbf{A} - \mathbf{B})$ across all rank-$k$ matrices $\mathbf{B}$, for some loss function $\mathcal{L}$. The rank parameter $k$ should be chosen to accurately represent the complexity of the underlying model chosen to fit the data, and thus the low-rank approximation problem is often used for mathematical modeling and data compression.

Similarly, in the classic column subset selection problem, the goal is to choose $k$ columns $\mathbf{U}$ of $\mathbf{A}$ so as to minimize $\mathcal{L}(\mathbf{A} - \mathbf{U}\mathbf{V})$ across all choices of $\mathbf{V} \in \mathbb{R}^{k \times d}$. Although low-rank approximation can reveal important latent structure among the dataset, the resulting linear combinations may not be as interpretable as simply selecting $k$ features. The column subset selection problem is therefore

38th Conference on Neural Information Processing Systems (NeurIPS 2024).

a version of low-rank approximation with the restriction that the left factor must be $k$ columns of the data matrix. Column subset selection also tends to result in sparse models. For example, if the columns of $\mathbf{A}$ are sparse, then the columns in the left factor $\mathbf{U}$ are also sparse. Thus in some cases, column subset selection, often also called feature selection, can be more useful than general low-rank approximation.

**Algorithmic fairness.** Unfortunately, real-world machine learning algorithms across a wide variety of domains have recently produced a number of undesirable outcomes from the lens of generalization. For example, [BS16] noted that decision-making processes using data collected from smartphone devices reporting poor road quality could potentially underserve poorer communities with less smartphone ownership. [KMM15] observed that search queries for CEOs overwhelmingly returned images of white men, while [BG18] observed that facial recognition software exhibited different accuracy rates for white men compared with dark-skinned women.

Initial attempts to explain these issues can largely be categorized into either "biased data" or "biased algorithms", where the former might include training data that is significantly misrepresenting the true statistics of some sub-population, while the latter might sacrifice accuracy on a specific sub-population in order to achieve better global accuracy. As a result, an increasingly relevant line of active work has focused on designing *fair* algorithms. An immediate challenge is to formally define the desiderata demanded from fair algorithmic design and indeed multiple natural quantitative measures of fairness have been proposed [HPS16, Cho17, KMR17, BHJ$^+$21]. However, [KMR17, CPF$^+$17] showed that many of these conditions for fairness cannot be simultaneously achieved.

In this paper, we focus on *socially fair* algorithms, which seek to optimize the performance of the algorithm across all sub-populations. That is, for the purposes of low-rank approximation and column subset selection, the goal is to minimize the maximum cost across all sub-populations. For socially fair low-rank approximation, the input is a set of data matrices $\mathbf{A}^{(1)} \in \mathbb{R}^{n_1 \times d}, \dots, \mathbf{A}^{(\ell)} \in \mathbb{R}^{n_\ell \times d}$ corresponding to $\ell$ groups and a rank parameter $k$, and the goal is to determine a set of $k$ factors $\mathbf{U}_1, \dots, \mathbf{U}_k \in \mathbb{R}^d$ that span matrices $\mathbf{B}^{(1)}, \dots, \mathbf{B}^{(\ell)}$, which minimize $\max_{i \in [\ell]} \|\mathbf{A}^{(i)} - \mathbf{B}^{(i)}\|_F$. Due to the Eckart–Young–Mirsky theorem stating that the Frobenius loss is minimized when each $\mathbf{A}^{(i)}$ is projected onto the span of $\mathbf{U} = \mathbf{U}_1 \circ \dots \circ \mathbf{U}_k$, the problem is equivalent to $\min_{\mathbf{U} \in \mathbb{R}^{k \times d}} \max_{i \in [\ell]} \|\mathbf{A}^{(i)} \mathbf{U}^\dagger \mathbf{U} - \mathbf{A}^{(i)}\|_F$, where $\dagger$ denotes the Moore-Penrose pseudoinverse. We remark that as we can scale the columns in each group (even each individual column), our formulation captures min-max normalized cost relative to the total Frobenius norm of each group and the optimal rank-$k$ reconstruction loss for each group.

## 1.1 Our Contributions and Technical Overview

In this paper, we study socially fair low-rank approximation and socially fair column subset selection.

**Fair low-rank approximation.** We first describe our results for socially fair low-rank approximation. We first show that under the assumption that $\text{P} \neq \text{NP}$, fair low-rank approximation cannot be approximated within any constant factor in polynomial time.

**Theorem 1.1.** *Fair low-rank approximation is NP-hard to approximate within any constant factor.*

We show Theorem 1.1 by reducing to the problem of minimizing the distance of a set of $n$ points in $d$-dimensional Euclidean space to all sets of $k$ dimensional linear subspaces, which was shown by [BGK00, DTV11] to be NP-hard to approximate within any constant factor. In fact, [BGK00, DTV11] showed that a constant-factor approximation to this problem requires runtime exponential in $k$ under a stronger assumption, the exponential time hypothesis [IP01]. We show similar results for the fair low-rank approximation problem.

**Theorem 1.2.** *Under the exponential time hypothesis, the fair low-rank approximation requires $2^{k^{\Omega(1)}}$ time to approximate within any constant factor.*

Together, Theorem 1.1 and Theorem 1.2 show that under standard complexity assumptions, we cannot achieve a constant-factor approximation to fair low-rank approximation using time polynomial in $n$ and exponential in $k$. We thus consider additional relaxations, such as bicriteria approximation (Theorem 1.4) or $2^{\text{poly}(k)}$ runtime (Theorem 1.3). On the positive side, we first show that for a constant number of groups and constant-factor accuracy, it suffices to use runtime $2^{\text{poly}(k)}$ rather

than the naïve $n^{\mathrm{poly}(k)}$, which is a substantial improvement when the dataset has a large number of observations, i.e., $n$ is large.

**Theorem 1.3.** *Given an accuracy parameter $\varepsilon \in (0,1)$, there exists an algorithm which outputs $\widetilde{\mathbf{V}} \in \mathbb{R}^{k \times d}$ such that with probability at least $\frac{2}{3}$, $\max_{i \in [\ell]} \|\mathbf{A}^{(i)}(\widetilde{\mathbf{V}})^{\dagger}\widetilde{\mathbf{V}} - \mathbf{A}^{(i)}\|_F \leq (1 + \varepsilon) \cdot \min_{\mathbf{V} \in \mathbb{R}^{k \times d}} \max_{i \in [\ell]} \|\mathbf{A}^{(i)}\mathbf{V}^{\dagger}\mathbf{V} - \mathbf{A}^{(i)}\|_F$. The algorithm uses runtime $\frac{1}{\varepsilon} \mathrm{poly}(n) \cdot (2\ell)^{\mathcal{O}(N)}$, for $n = \sum_{i=1}^{\ell} n_i$ and $N = \mathrm{poly}\left(\ell, k, \frac{1}{\varepsilon}\right)$.*

Next, we show that there exists a bicriteria approximation algorithm for fair low-rank approximation that uses polynomial runtime.

**Theorem 1.4.** *Given a trade-off parameter $c \in (0,1)$, there exists an algorithm that outputs $\widetilde{\mathbf{V}} \in \mathbb{R}^{t \times d}$ for $t = \mathcal{O}\left(k(\log \log k)(\log^2 d)\right)$ such that with probability at least $\frac{2}{3}$,*

$$\max_{i \in [\ell]} \|\mathbf{A}^{(i)}(\widetilde{\mathbf{V}})^{\dagger}\widetilde{\mathbf{V}} - \mathbf{A}^{(i)}\|_F \leq \ell^c \cdot 2^{1/c} \cdot \mathcal{O}\left(k(\log \log k)(\log d)\right) \cdot \min_{\mathbf{V} \in \mathbb{R}^{k \times d}} \max_{i \in [\ell]} \|\mathbf{A}^{(i)}\mathbf{V}^{\dagger}\mathbf{V} - \mathbf{A}^{(i)}\|_F.$$

*The algorithm uses runtime polynomial in $n$ and $d$.*

The algorithm for Theorem 1.4 substantially differs from that of Theorem 1.3. For one, we can no longer use a polynomial system solver, because it would be infeasible to achieve polynomial runtime. Instead, we observe that for sufficiently large $p$, we have $\max \|\mathbf{x}\|_{\infty} = (1 \pm \varepsilon)\|\mathbf{x}\|_p$ and thus focus on optimizing $\min_{\mathbf{V} \in \mathbb{R}^{k \times d}} \left(\sum_{i \in [\ell]} \|\mathbf{A}^{(i)}\mathbf{V}^{\dagger}\mathbf{V} - \mathbf{A}^{(i)}\|_F^p\right)^{1/p}$. However, the terms $\|\mathbf{A}^{(i)}\mathbf{V}^{\dagger}\mathbf{V} - \mathbf{A}^{(i)}\|_F^p$ are difficult to handle, so we apply Dvoretzky's Theorem, i.e., Theorem 3.2, to generate matrices $\mathbf{G}$ and $\mathbf{H}$ so that $(1 - \varepsilon)\|\mathbf{GMH}\|_p \leq \|\mathbf{M}\|_F \leq (1 + \varepsilon)\|\mathbf{GMH}\|_p$, for all matrices $\mathbf{M} \in \mathbb{R}^{n \times d}$, so that it suffices to approximately solve $\min_{\mathbf{X} \in \mathbb{R}^{k \times d}} \|\mathbf{GAHSX} - \mathbf{GAH}\|_p$, for $\mathbf{A} = \mathbf{A}^{(1)} \circ \ldots \circ \mathbf{A}^{(\ell)}$.

Although low-rank approximation with $L_p$ loss cannot be well-approximated in polynomial time, we recall that there exists a matrix $\mathbf{S}$ that samples a "small" number of columns of $\mathbf{A}$ to provide a coarse bicriteria approximation to $L_p$ low-rank approximation [WY23]. However, we require a solution with dimension $d$ and thus we seek to solve the regression problem $\min_{\mathbf{X}} \|\mathbf{GAHSX} - \mathbf{GAH}\|_p$. Thus, we consider a Lewis weight sampling matrix $\mathbf{T}$ such that

$$\frac{1}{2}\|\mathbf{TGAHSX} - \mathbf{TGAH}\|_p \leq \|\mathbf{GAHSX} - \mathbf{GAH}\|_p \leq 2\|\mathbf{TGAHSX} - \mathbf{TGAH}\|_p.$$

and again note that $(\mathbf{TGAHS})^{\dagger}\mathbf{TGAH}$ is the closed-form solution to the minimization problem $\min_{\mathbf{x}} \|\mathbf{TGAHSX} - \mathbf{TGAH}\|_F$, which only provides a small distortion to the $L_p$ regression problem, since $\mathbf{TGAH}$ has a small number of rows due to the dimensionality reduction. We then observe that by Dvoretzky's Theorem, $(\mathbf{TGAHS})^{\dagger}\mathbf{TGA}$ is a "good" approximate solution to the original fair low-rank approximation problem. Given $\delta \in (0,1)$, the success probabilities for both Theorem 1.3 and Theorem 1.4 can be boosted to arbitrary $1 - \delta$ by taking the minimum of $\mathcal{O}\left(\log \frac{1}{\delta}\right)$ independent instances of the algorithm, at the cost of increasing the runtime by the same factor.

**Fair column subset selection.** We next describe our results for fair column subset selection. We give a bicriteria approximation algorithm for fair column subset selection that uses polynomial runtime.

**Theorem 1.5.** *Given input matrices $\mathbf{A}^{(i)} \in \mathbb{R}^{n_i \times d}$ with $n = \sum n_i$, there exists an algorithm that selects a set $S$ of $k' = \mathcal{O}(k \log k)$ columns such that with probability at least $\frac{2}{3}$, $S$ is a $\mathcal{O}(k(\log \log k)(\log d))$-approximation to the fair column subset selection problem. The algorithm uses runtime polynomial in $n$ and $d$.*

The immediate challenge in adapting the previous approach for fair low-rank approximation to fair column subset selection is that we required the Gaussian matrices $\mathbf{G}, \mathbf{H}$ to embed the awkward maximum of Frobenius losses $\min \max_{i \in [\ell]} \|\cdot\|_F$ into the more manageable $L_p$ loss $\min \|\cdot\|_p$ through $\mathbf{GAH}$. However, selecting columns of $\mathbf{GAH}$ does not correspond to selecting columns of the input matrices $\mathbf{A}^{(1)}, \ldots, \mathbf{A}^{(\ell)}$.

Instead, we view the bicriteria solution $\widetilde{\mathbf{V}}$ from fair low-rank approximation as a good starting point for the right factor for fair low-rank approximation. Thus we consider the multi-response regression problem $\max_{i \in [\ell]} \min_{\mathbf{B}^{(i)}} \|\mathbf{B}^{(i)}\widetilde{\mathbf{V}} - \mathbf{A}^{(i)}\|_F$. We then argue through Dvoretzky's theorem

that a leverage score sampling matrix $\mathbf{S}$ that samples $\mathcal{O}\left(k \log k\right)$ columns of $\widetilde{\mathbf{V}}$ will provide a good approximation to the column subset selection problem. We defer the formal exposition to Section 4.

**Empirical evaluations.** Finally, in Section 5, we perform a number of experimental results on socially fair low-rank approximation, comparing the performance of the socially fair low-rank objective values associated with the outputs of the bicriteria fair low-rank approximation algorithm and the standard (non-fair) low-rank approximation that outputs the top $k$ right singular vectors of the singular value decomposition.

Our experiments are on the Default of Credit Card Clients dataset [YL09], which is a common human-centric data used for benchmarks on fairness, e.g., see [STM+18]. We perform empirical evaluations comparing the objective value and the runtime of our bicriteria algorithm with the aforementioned baseline, across various subsample sizes and rank parameters. Our results demonstrate that our bicriteria algorithm can perform better than the standard low-rank approximation algorithm across various parameter settings, even when the bicriteria algorithm is not allowed a larger rank than the baseline. Moreover, we show that our algorithm is quite efficient and in fact, the final step of extracting the low-rank factors is faster than the singular value decomposition baseline due to a smaller input matrix. All in all, our empirical evaluations indicate that our bicriteria algorithm can perform well in practice, thereby reinforcing the theoretical guarantees of the algorithm. Finally, we give a number of additional experiments on synthetic datasets, in Appendix D.

**Paper organization.** We first described a number of related works in Section 2. We detail our socially fair low-rank approximation algorithms in Section 3 and our socially fair column subset selection algorithms in Section 4. We defer all proofs to the supplementary material. We present our experiments in Section 5 and Appendix D. The reader may also find it helpful to consult Appendix A for standard notation used in our paper and additional preliminaries.

## 2 Related Work

Initial insight into *socially fair data summarization* methods were presented by [STM+18], where the concept of *fair PCA* was explored. This study introduced the fairness metric of average reconstruction loss, expressed by the loss function $\text{loss}(\mathbf{A}, \mathbf{B}) := \|\mathbf{A} - \mathbf{B}\|_F^2 - \|\mathbf{A} - \mathbf{A}_k\|_F^2$, aiming to identify a $k$-dimensional subspace that minimizes the loss across the groups, with $\mathbf{A}_k$ representing the best rank-$k$ approximation of $\mathbf{A}$. Their proposed approach, in a two-group scenario, identifies a fair PCA of up to $k + 1$ dimensions that is not worse than the optimal fair PCA with $k$ dimensions. When extended to $\ell$ groups, this method requires an additional $k + \ell - 1$ dimensions. Subsequently, [TSS+19] explored fair PCA from a distinct objective perspective, seeking a projection matrix $\mathbf{P}$ optimizing $\min_{i \in [\ell]} \|\mathbf{A}^{(i)} \mathbf{P}\|_F^2$. A pivotal difference between these works and ours is our focus on the reconstruction error objective, a widely accepted objective for regression and low-rank approximation tasks. Alternatively, [OA19, LKO+22, KDRZ23] explored a different formulation of fair PCA. The main objective is to ensure that data representations are not influenced by demographic attributes. In particular, when a classifier is exposed only to the projection of points onto the $k$-dimensional subspace, it should be unable to predict the demographic attributes.

Recently, [MOT24] studied a fair column subset selection objective similar to ours, focusing on the setting with two groups (i.e., $\ell = 2$). They established the problem's NP-hardness and introduced a polynomial-time solution that offers relative-error guarantees while selecting a column subset of size $\mathcal{O}\left(k\right)$.

For fair regression, initial research focused on designing models that offer similar treatment to instances with comparable observed results by incorporating *fairness regularizers* [BHJ+17]. However, in [ADW19], the authors studied a fairness notion closer to our optimization problem, termed as "bounded group loss". In their work, the aim is to cap each group's loss within a specific limit while also optimizing the cumulative loss. Notably, their approach diverged from ours, with a focus on the sample complexity and the problem's generalization error bounds.

[AAK+22] studied a similar socially fair regression problem under the name min-max regression. In their setting, the goal is to minimize the maximum loss over a mixture distribution, given samples from the mixture; our fair regression setting can be reduced to theirs. [AAK+22] observed that a maximum of norms is a convex function and can therefore be solved using projected stochastic gradient descent.

The term "socially fair" was first introduced in the context of clustering, aiming to optimize clustering costs across predefined group sets [GSV21, ABV21]. In subsequent studies, tight approximation algorithms [MV21, CMV22], FPT approaches [GJ23], and bicriteria approximation algorithms [GSV22] for socially fair clustering have been presented.

## 3   Socially Fair Low-Rank Approximation

In this section, we consider algorithms and hardness for socially fair low-rank approximation. Let $n_1, \ldots, n_\ell$ be positive integers and for each $i \in [\ell]$, let $\mathbf{A}^{(i)} \in \mathbb{R}^{n_i \times d}$. Then for a norm $\| \cdot \|$, we define the fair low-rank approximation problem to be $\min_{\mathbf{V} \in \mathbb{R}^{k \times d}} \max_{i \in [\ell]} \|\mathbf{A}^{(i)} \mathbf{V}^\dagger \mathbf{V} - \mathbf{A}^{(i)}\|$.

### 3.1   $(1 + \varepsilon)$-Approximation Algorithm for Fair Low-Rank Approximation

We first give a $(1 + \varepsilon)$-approximation algorithm for fair low-rank approximation that uses runtime $\frac{1}{\varepsilon} \operatorname{poly}(n) \cdot (2\ell)^{\mathcal{O}(N)}$, for $n = \sum_{i=1}^\ell n_i$ and $N = \operatorname{poly}\left(\ell, k, \frac{1}{\varepsilon}\right)$.

The algorithm first finds a value $\alpha$ that is an $\ell$-approximation to the optimal solution, i.e., $\min_{\mathbf{V} \in \mathbb{R}^{k \times d}} \max_{i \in [\ell]} \|\mathbf{A}^{(i)} \mathbf{V}^\dagger \mathbf{V} - \mathbf{A}^{(i)}\|_F$ is at most $\alpha \le \ell \cdot \min_{\mathbf{V} \in \mathbb{R}^{k \times d}} \max_{i \in [\ell]} \|\mathbf{A}^{(i)} \mathbf{V}^\dagger \mathbf{V} - \mathbf{A}^{(i)}\|_F$. We then repeatedly decrease $\alpha$ by $(1 + \varepsilon)$ while checking if the resulting quantity is still achievable. To efficiently check if $\alpha$ is achievable, we first apply dimensionality reduction to each of the matrices by right-multiplying by an affine embedding matrix $\mathbf{S}$, so that

$$(1 - \varepsilon)\|\mathbf{A}^{(i)} \mathbf{V}^\dagger \mathbf{V} - \mathbf{A}^{(i)}\|_F^2 \le \|\mathbf{A}^{(i)} \mathbf{V}^\dagger \mathbf{V} \mathbf{S} - \mathbf{A}^{(i)} \mathbf{S}\|_F^2 \le (1 + \varepsilon)\|\mathbf{A}^{(i)} \mathbf{V}^\dagger \mathbf{V} - \mathbf{A}^{(i)}\|_F^2,$$

for all rank $k$ matrices $\mathbf{V}$ and all $i \in [\ell]$.

Now if we knew $\mathbf{V}$, then for each $i \in [\ell]$, we can find $\mathbf{X}^{(i)}$ minimizing $\|\mathbf{X}^{(i)} \mathbf{V} \mathbf{S} - \mathbf{A}^{(i)} \mathbf{S}\|_F^2$ and the resulting quantity will approximate $\|\mathbf{A}^{(i)} \mathbf{V}^\dagger \mathbf{V} - \mathbf{A}^{(i)}\|_F^2$. In fact, we know that the minimizer is $(\mathbf{A}^{(i)} \mathbf{S})(\mathbf{V} \mathbf{S})^\dagger$ through the closed form solution to the regression problem. Let $\mathbf{R}^{(i)}$ be defined so that $(\mathbf{A}^{(i)} \mathbf{S})(\mathbf{V} \mathbf{S})^\dagger \mathbf{R}^{(i)}$ has orthonormal columns, so that

$$\|(\mathbf{A}^{(i)} \mathbf{S})(\mathbf{V} \mathbf{S})^\dagger \mathbf{R}^{(i)})((\mathbf{A}^{(i)} \mathbf{S})(\mathbf{V} \mathbf{S})^\dagger \mathbf{R}^{(i)})^\dagger \mathbf{A}^{(i)} \mathbf{S} - \mathbf{A}^{(i)} \mathbf{S}\|_F^2 = \min_{\mathbf{X}^{(i)}} \|\mathbf{X}^{(i)} \mathbf{V} \mathbf{S} - \mathbf{A}^{(i)} \mathbf{S}\|_F^2,$$

and so we require that if $\alpha$ is feasible, then $\alpha \ge \|(\mathbf{A}^{(i)} \mathbf{S})(\mathbf{V} \mathbf{S})^\dagger \mathbf{R}^{(i)})((\mathbf{A}^{(i)} \mathbf{S})(\mathbf{V} \mathbf{S})^\dagger \mathbf{R}^{(i)})^\dagger \mathbf{A}^{(i)} \mathbf{S} - \mathbf{A}^{(i)} \mathbf{S}\|_F^2$. Unfortunately, we do not know $\mathbf{V}$, so instead we use a polynomial solver to check whether there exists such a $\mathbf{V}$. We remark that similar guessing strategies were employed by [RSW16, KPRW19, BWZ19, VVWZ23] and in particular, [RSW16] also uses a polynomial system in conjunction with the guessing strategy. Thus we write $\mathbf{Y} = \mathbf{V} \mathbf{S}$ and its pseudoinverse $\mathbf{W} = (\mathbf{V} \mathbf{S})^\dagger$ and check whether there exists a satisfying assignment to the above inequality, given the constraints (1) $\mathbf{Y} \mathbf{W} \mathbf{Y} = \mathbf{Y}$, (2) $\mathbf{W} \mathbf{Y} \mathbf{W} = \mathbf{W}$, and (3) $\mathbf{A}^{(i)} \mathbf{S} \mathbf{W} \mathbf{R}^{(i)}$ has orthonormal columns. Note that since $\mathbf{V} \in \mathbb{R}^{k \times d}$, then implementing the polynomial solver naïvely could require $kd$ variables and thus use $2^{\Omega(dk)}$ runtime. Instead, we note that we only work with $\mathbf{V} \mathbf{S}$, which has dimension $k \times m$ for $m = \mathcal{O}\left(\frac{k^2}{\varepsilon^2} \log \ell\right)$, so that the polynomial solver only uses $2^{\operatorname{poly}(mk)}$ time.

We now show a crucial structural property that allows us to distinguish between the case where a guess $\alpha$ for the optimal value $\mathsf{OPT}$ exceeds $(1 + \varepsilon)\mathsf{OPT}$ or is smaller than $(1 - \varepsilon)\mathsf{OPT}$ by simply looking at a polynomial system solver on an affine embedding.

**Lemma 3.1.** *Let $\mathbf{V} \in \mathbb{R}^{k \times d}$ be the optimal solution to the fair low-rank approximation problem for inputs $\mathbf{A}^{(1)}, \ldots, \mathbf{A}^{(\ell)}$, where $\mathbf{A}^{(i)} \in \mathbb{R}^{n_i \times d}$, and suppose $\mathsf{OPT} = \max_{i \in [\ell]} \|\mathbf{A}^{(i)} \mathbf{V}^\dagger \mathbf{V} - \mathbf{A}^{(i)}\|_F^2$. Let $\mathbf{S}$ be an affine embedding for $\mathbf{V}$ and let $\mathbf{W} = (\mathbf{V} \mathbf{S})^\dagger \in \mathbb{R}^{k \times m}$. For $i \in [\ell]$, let $\mathbf{Z}^{(i)} = \mathbf{A}^{(i)} \mathbf{S} \mathbf{W} \in \mathbb{R}^{n_i \times k}$ and $\mathbf{R}^{(i)} \in \mathbb{R}^{k \times k}$ be defined so that $\mathbf{A}^{(i)} \mathbf{S} \mathbf{W} \mathbf{R}^{(i)}$ has orthonormal columns. If $\alpha \ge (1 + \varepsilon) \cdot \mathsf{OPT}$, then for each $i \in [\ell]$, $\alpha \ge \|(\mathbf{A}^{(i)} \mathbf{S} \mathbf{W} \mathbf{R}^{(i)})(\mathbf{A}^{(i)} \mathbf{S} \mathbf{W} \mathbf{R}^{(i)})^\dagger \mathbf{A}^{(i)} - \mathbf{A}^{(i)}\|_F^2$. If $\alpha < (1 - \varepsilon) \cdot \mathsf{OPT}$, then there exists $i \in [\ell]$, such that $\alpha < \|(\mathbf{A}^{(i)} \mathbf{S} \mathbf{W} \mathbf{R}^{(i)})(\mathbf{A}^{(i)} \mathbf{S} \mathbf{W} \mathbf{R}^{(i)})^\dagger \mathbf{A}^{(i)} - \mathbf{A}^{(i)}\|_F^2$.*

**Algorithm 1** Input to polynomial solver

**Input:** $\mathbf{A}^{(1)}, \ldots, \mathbf{A}^{(\ell)}, \mathbf{S}, \alpha$
**Output:** Feasibility of polynomial system
 1: **Polynomial variables**
 2: Let $\mathbf{Y} = (\mathbf{VS}) \in \mathbb{R}^{k \times m}$ be $mk$ variables
 3: Let $\mathbf{W} = (\mathbf{VS})^\dagger \in \mathbb{R}^{m \times k}$ be $mk$ variables

 4: Let $\mathbf{R}^{(i)} \in \mathbb{R}^{k \times k}$ for each $i \in [\ell]$ be $\ell k^2$ variables
 5: **System constraints**
 6: $\mathbf{YWY} = \mathbf{Y}, \mathbf{WYW} = \mathbf{W}$
 7: $\mathbf{A}^{(i)}\mathbf{SWR}^{(i)}$ has orthonormal columns
 8: $\alpha \geq \|(\mathbf{A}^{(i)}\mathbf{SWR}^{(i)})(\mathbf{A}^{(i)}\mathbf{SWR}^{(i)})^\dagger \mathbf{A}^{(i)} - \mathbf{A}^{(i)}\|_F^2$
 9: **Run polynomial system solver**
10: If feasible, output $\mathbf{V} = (\mathbf{A}^{(1)}\mathbf{SWR}^{(1)})^\dagger \mathbf{A}^{(1)}$. Otherwise, output $\perp$.

**Algorithm 2** $(1+\varepsilon)$-approximation for fair low-rank approximation

**Input:** $\mathbf{A}^{(i)} \in \mathbb{R}^{n_i \times d}$ for all $i \in [\ell]$, rank parameter $k > 0$, accuracy parameter $\varepsilon \in (0, 1)$
**Output:** $(1 + \varepsilon)$-approximation for fair LRA
 1: Let $\alpha$ be an $\ell$-approximation for the fair LRA problem
 2: Let $\mathbf{S}$ be generated from a random affine embedding distribution
 3: **while** Algorithm 1 on input $\mathbf{A}^{(1)}, \ldots, \mathbf{A}^{(\ell)}$, $\mathbf{S}$, and $\alpha$ does not return $\perp$ **do**
 4:     Let $\mathbf{V}$ be the output of Algorithm 1 on input $\mathbf{A}^{(1)}, \ldots, \mathbf{A}^{(\ell)}$, $\mathbf{S}$, and $\alpha$
 5:     $\alpha \leftarrow \frac{\alpha}{1+\varepsilon}$
 6: **end while**
 7: Return $\mathbf{V}$

## 3.2 Bicriteria Algorithm

To achieve polynomial time for our bicriteria algorithm, we can no longer use a polynomial system solver. Instead, we observe that for sufficiently large $p$, we have $\max \|\mathbf{x}\|_\infty = (1 \pm \varepsilon)\|\mathbf{x}\|_p$. Thus, in place of optimizing $\min_{\mathbf{V} \in \mathbb{R}^{k \times d}} \max_{i \in [\ell]} \|\mathbf{A}^{(i)}\mathbf{V}^\dagger \mathbf{V} - \mathbf{A}^{(i)}\|_F$, we instead optimize $\min_{\mathbf{V} \in \mathbb{R}^{k \times d}} \left(\sum_{i \in [\ell]} \|\mathbf{A}^{(i)}\mathbf{V}^\dagger \mathbf{V} - \mathbf{A}^{(i)}\|_F^p\right)^{1/p}$. However, the terms $\|\mathbf{A}^{(i)}\mathbf{V}^\dagger \mathbf{V} - \mathbf{A}^{(i)}\|_F^p$ are unwieldy to work with. Thus we instead use Dvoretzky's Theorem, i.e., Theorem 3.2, to embed $L_2$ into $L_p$, by generating matrices $\mathbf{G}$ and $\mathbf{H}$ so that $(1 - \varepsilon)\|\mathbf{GMH}\|_p \leq \|\mathbf{M}\|_F \leq (1 + \varepsilon)\|\mathbf{GMH}\|_p$, for all matrices $\mathbf{M} \in \mathbb{R}^{n \times d}$.

Now, writing $\mathbf{A} = \mathbf{A}^{(1)} \circ \ldots \circ \mathbf{A}^{(\ell)}$, it suffices to approximately solve $\min_{\mathbf{X} \in \mathbb{R}^{k \times d}} \|\mathbf{GAHSX} - \mathbf{GAH}\|_p$. Unfortunately, low-rank approximation with $L_p$ loss still cannot be approximated to $(1 + \varepsilon)$-factor in polynomial time, and in fact $\mathbf{GAH}$ has dimension $n' \times d'$ with $n' \geq n$ and $d' \geq d$. Hence, we first apply dimensionality reduction by appealing to a result of [WY23] showing that there exists a matrix $\mathbf{S}$ that samples a "small" number of columns of $\mathbf{A}$ to provide a coarse bicriteria approximation to $L_p$ low-rank approximation. Now to lift the solution back to dimension $d$, we would like to solve regression problem $\min_{\mathbf{X}} \|\mathbf{GAHSX} - \mathbf{GAH}\|_p$. To that end, we consider a Lewis weight sampling matrix $\mathbf{T}$ such that

$$\frac{1}{2}\|\mathbf{TGAHSX} - \mathbf{TGAH}\|_p \leq \|\mathbf{GAHSX} - \mathbf{GAH}\|_p \leq 2\|\mathbf{TGAHSX} - \mathbf{TGAH}\|_p.$$

We then note that $(\mathbf{TGAHS})^\dagger \mathbf{TGAH}$ is the minimizer of the problem $\min_{\mathbf{x}} \|\mathbf{TGAHSX} - \mathbf{TGAH}\|_F$, which only provides a small distortion to the $L_p$ regression problem, since $\mathbf{TGAH}$ has a small number of rows due to the dimensionality reduction. By Dvoretzky's Theorem, we have that $(\mathbf{TGAHS})^\dagger \mathbf{TGA}$ is a "good" approximate solution to the original fair low-rank approximation problem. The algorithm appears in full in Algorithm 3.

**Algorithm 3** Bicriteria approximation for fair low-rank approximation

**Input:** $\mathbf{A}^{(i)} \in \mathbb{R}^{n_i \times d}$ for all $i \in [\ell]$, rank parameter $k > 0$, trade-off parameter $c \in (0, 1)$
**Output:** Bicriteria approximation for fair LRA
 1: Generate Gaussian matrices $\mathbf{G} \in \mathbb{R}^{n' \times n}, \mathbf{H} \in \mathbb{R}^{d \times d'}$ through Theorem 3.2
 2: Let $\mathbf{S} \in \mathbb{R}^{n' \times t}, \mathbf{Z} \in \mathbb{R}^{t \times d'}$ be the output of Theorem 3.3 on input $\mathbf{GAH}$
 3: Let $\mathbf{T} \in \mathbb{R}^{s \times n'}$ be a Lewis weight sampling matrix for $\mathbf{GAHSX} - \mathbf{GAH}$
 4: Let $\widetilde{\mathbf{V}} \leftarrow (\mathbf{TGAHS})^\dagger (\mathbf{TGA})$
 5: Return $\widetilde{\mathbf{V}}$

We use the following notion of Dvoretzky's theorem to embed the problem into entrywise $L_p$ loss.

**Theorem 3.2** (Dvoretzky's Theorem, e.g., Theorem 1.2 in [PVZ17], Fact 15 in [SW18]). *Let $p \geq 1$ be a parameter and let*

$$m \gtrsim m(n, p, \varepsilon) = \begin{cases} \frac{p^p n}{\varepsilon^2}, & \varepsilon \leq (Cp)^{\frac{p}{2}} n^{-\frac{p-2}{2(p-1)}} \\ \frac{(np)^{p/2}}{\varepsilon}, & \varepsilon \in \left( (Cp)^{\frac{p}{2}} n^{-\frac{p-2}{2(p-1)}}, \frac{1}{p} \right] \\ \frac{n^{p/2}}{p^{p/2}\varepsilon^{p/2}} \log^{p/2} \frac{1}{\varepsilon}, & \frac{1}{p} < \varepsilon < 1. \end{cases}$$

*Then there exists a family $\mathcal{G}$ of random scaled Gaussian matrices with dimension $\mathbb{R}^{m \times n}$ such that for $G \sim \mathcal{G}$, with probability at least $1 - \delta$, simultaneously for all $\mathbf{y} \in \mathbb{R}^n$, $(1 - \varepsilon)\|\mathbf{y}\|_2 \leq \|\mathbf{Gy}\|_p \leq (1 + \varepsilon)\|\mathbf{y}\|_2$.*

We use the following algorithm from [WY23] to perform dimensionality reduction so that switching between $L_2$ and $L_p$ loss will incur smaller error. See also [CGK+17].

**Theorem 3.3** (Theorem 1.5 in [WY23]). *Let $\mathbf{A} \in \mathbb{R}^{n \times d}$ and let $k \geq 1$. Let $s = \mathcal{O}(k \log \log k)$. Then there exists a polynomial-time algorithm that outputs a matrix $\mathbf{S} \in \mathbb{R}^{d \times t}$ that samples $t = \mathcal{O}\left(k(\log \log k)(\log^2 d)\right)$ columns of $\mathbf{A}$ and a matrix $\mathbf{Z} \in \mathbb{R}^{t \times d}$ such that $\|\mathbf{A} - \mathbf{ASZ}\|_p \leq 2^p \cdot \mathcal{O}(\sqrt{s}) \cdot \min_{\mathbf{U} \in \mathbb{R}^{n \times k}, \mathbf{V} \in \mathbb{R}^{k \times d}} \|\mathbf{A} - \mathbf{UV}\|_p$.*

We recall the following construction to use Lewis weights to achieve an $L_p$ subspace embedding.

**Theorem 3.4** ([CP15]). *Let $\varepsilon \in (0, 1)$ and $p \geq 2$. Let $\mathbf{A} \in \mathbb{R}^{n \times d}$ and $s = \mathcal{O}\left(d^{p/2} \log d\right)$. Then there exists a polynomial-time algorithm that outputs a matrix $\mathbf{S} \in \mathbb{R}^{s \times n}$ that samples and reweights $s$ rows of $\mathbf{A}$, such that with probability at least $0.99$, simultaneously for all $\mathbf{x} \in \mathbb{R}^d$, $(1 - \varepsilon)\|\mathbf{Ax}\|_p^p \leq \|\mathbf{SAx}\|_p^p \leq (1 + \varepsilon)\|\mathbf{Ax}\|_p^p$.*

We then show that Algorithm 3 provides a bicriteria approximation.

**Lemma 3.5.** *Let $\widetilde{\mathbf{V}}$ be the output of Algorithm 3. Then with probability at least $\frac{9}{10}$, $\max_{i \in [\ell]} \|\mathbf{A}^{(i)}(\widetilde{\mathbf{V}})^\dagger \widetilde{\mathbf{V}} - \mathbf{A}^{(i)}\|_F$ is at most $\ell^c \cdot 2^{1/c} \cdot \mathcal{O}(k(\log \log k)(\log d)) \max_{i \in [\ell]} \|\mathbf{A}^{(i)}(\widetilde{\mathbf{V}})^\dagger \widetilde{\mathbf{V}} - \mathbf{A}^{(i)}\|_F$, where $c$ is the trade-off parameter input.*

Since the generation of Gaussian matrices and the Lewis weight sampling matrix both only require polynomial time, it follows that our algorithm uses polynomial time overall. Hence, we have Theorem 1.4.

# 4 Socially Fair Column Subset Selection

In this section, we consider socially fair column subset selection, where the goal is to identify a matrix $\mathbf{C} \in \mathbb{R}^{d \times k}$ that selects $k$ columns to minimize $\min_{\mathbf{C} \in \mathbb{R}^{d \times k}, \|\mathbf{C}\|_0 \leq k, \mathbf{B}^{(i)}} \max_{i \in [\ell]} \|\mathbf{A}^{(i)}\mathbf{CB}^{(i)} - \mathbf{A}^{(i)}\|_F$.

---

**Algorithm 4** Bicriteria approximation for fair column subset selection

---

**Input:** $\mathbf{A}^{(i)} \in \mathbb{R}^{n_i \times d}$ for all $i \in [\ell]$, rank parameter $k > 0$, trade-off parameter $c \in (0, 1)$
**Output:** Bicriteria approximation for fair column subset selection
 1: Acquire $\widetilde{\mathbf{V}}$ from Algorithm 3
 2: Generate Gaussian $\mathbf{G} \in \mathbb{R}^{n' \times n}$ through Theorem 3.2
 3: Let $\mathbf{S} \in \mathbb{R}^{d \times k'}$ be a leverage score sampling matrix that samples $k' = \mathcal{O}(k \log k)$ columns of $\widetilde{\mathbf{V}}$

 4: $\mathbf{M}^{(i)} = \mathbf{S}^\dagger (\widetilde{\mathbf{V}})^\dagger \widetilde{\mathbf{V}}$ for all $i \in [\ell]$
 5: Return $\mathbf{A}^{(i)}\mathbf{S}, \{\mathbf{M}^{(i)}\}$

---

We first provide preliminaries on leverage score sampling.

**Definition 4.1.** *Given a matrix $\mathbf{M} \in \mathbb{R}^{n \times d}$, we define the leverage score $\sigma_i$ of each row $\mathbf{m}_i$ with $i \in [n]$ by $\mathbf{m}_i(\mathbf{M}^\dagger \mathbf{M})^{-1}\mathbf{m}_i^\top$. Equivalently, for the singular value decomposition $\mathbf{M} = \mathbf{U\Sigma V}$, the leverage score of row $\mathbf{m}_i$ is also the squared row norm of $\mathbf{u}_i$.*

We recall in Appendix C that it can be shown the sum of the leverage scores for an input matrix $\mathbf{M} \in \mathbb{R}^{n \times d}$ is upper bounded by $d$ and moreover, given the leverage scores of $\mathbf{M}$, it suffices to sample only $\mathcal{O}(d \log n)$ rows of $\mathbf{M}$ to achieve a constant factor subspace embedding of $\mathbf{M}$. Because the leverage scores of $\mathbf{M}$ can be computed directly from the singular value decomposition of $\mathbf{M}$, which can be computed in $\mathcal{O}(nd^\omega + dn^\omega)$ time where $\omega$ is the exponent of matrix multiplication, then the leverage scores of $\mathbf{M}$ can be computed in polynomial time.

Finally, we recall that to provide a constant factor approximation to $L_p$ regression, it suffices to compute a constant factor subspace embedding, e.g., through leverage score sampling. The proof is through the triangle inequality and is well-known among the active sampling literature [CP19, PPP21, MMWY22, MMM+22, MMM+23], e.g., a generalization of Lemma 2.1 in [MMM+22]. For completeness, we provide the proof in Appendix C.

**Lemma 4.2.** *Given a matrix* $\mathbf{M} \in \mathbb{R}^{n \times d}$*, let* $\mathbf{S}$ *be a matrix such that for all* $\mathbf{x} \in \mathbb{R}^d$ *and* $\mathbf{v} \in \mathbb{R}^n$*,*

$$\frac{11}{12}\|\mathbf{M}\mathbf{x}\|_2 \leq \|\mathbf{S}\mathbf{M}\mathbf{x}\|_2 \leq \frac{13}{12}\|\mathbf{M}\mathbf{x}\|_2, \ \mathbb{E}\left[\|\mathbf{S}\mathbf{v}\|_2^2\right] = \|\mathbf{v}\|_2^2.$$

*For a fixed* $\mathbf{B} \in \mathbb{R}^{n \times m}$ *where* $\mathbf{B} = \mathbf{b}_1 \circ \ldots \circ \mathbf{b}_m$ *with* $\mathbf{b}_i \in \mathbb{R}^n$ *for* $i \in [m]$*, let* $\widetilde{\mathbf{x}}_i = (\mathbf{S}\mathbf{M})^\dagger(\mathbf{S}\mathbf{b}_i)$*. Let* $\widetilde{\mathbf{X}} = \widetilde{\mathbf{x}_1} \circ \ldots \circ \widetilde{\mathbf{x}_m}$*. Then with probability at least* $0.97$*,* $\|\mathbf{M}\widetilde{\mathbf{X}} - \mathbf{B}\|_2 \leq 99 \min_{\mathbf{X}} \|\mathbf{M}\mathbf{X} - \mathbf{B}\|_2$*.*

We now give the correctness guarantees of Algorithm 4.

**Lemma 4.3.** *Let* $\mathbf{S}, \mathbf{M}^{(1)}, \ldots, \mathbf{M}^{(\ell)}$ *be the output of Algorithm 4. Then with probability at least* $0.8$*,* $\max_{i \in [\ell]} \|\mathbf{A}^{(i)}\mathbf{S}\mathbf{M}^{(i)} - \mathbf{A}^{(i)}\|_F$ *is at most* $\ell^c \cdot 2^{1/c} \cdot \mathcal{O}(k(\log \log k)(\log d)) \min_{\mathbf{V} \in \mathbb{R}^{k \times d}} \max_{i \in [\ell]} \|\mathbf{A}^{(i)}\mathbf{V}^\dagger\mathbf{V} - \mathbf{A}^{(i)}\|_F$*.*

We also have the following runtime guarantees on Algorithm 4.

**Lemma 4.4.** *The runtime of Algorithm 4 is* $\operatorname{poly}(n, d)$*.*

By combining Lemma 4.3 and Lemma 4.4, we have Theorem 1.5.

# 5 Empirical Evaluations

In this section, we describe our empirical evaluations for socially fair low-rank approximation on real-world datasets.

**Credit card dataset.** We used the Default of Credit Card Clients dataset [YL09], which has 30,000 observations across 23 features, including 17 numeric features. The dataset is a common human-centric data for experiments on fairness and was previously used as a benchmark by [STM+18] for studies on fair PCA. The study collected information from various customers including multiple previous payment statements, previous payment delays, and upcoming bill statements, as well as if the customer was able to pay the upcoming bill statement or not, i.e., defaulted on the bill statement. The dataset was accessed through the UCI repository [Yeh16].

**Experimental setup.** For the purposes of reproducibility, our empirical evaluations were conducted using Python 3.10 using a 64-bit operating system on an AMD Ryzen 7 5700U CPU, with 8GB RAM and 8 cores with base clock 1.80 GHz. We compare our bicriteria algorithm from Algorithm 3 against the standard non-fair low-rank approximation algorithm that outputs the top $k$ right singular vectors from the singular value decomposition. Gender was used as the sensitive attribute, so that all observations with one gender formed the matrix $\mathbf{A}^{(1)}$ and the other observations formed the matrix $\mathbf{A}^{(2)}$. As in Algorithm 3, we generate normalized Gaussian matrices $\mathbf{G}$ and $\mathbf{H}$ and then use $L_p$ Lewis weight sampling to generate a matrix $\mathbf{T}$. We generate matrices $\mathbf{T}$, $\mathbf{G}$, and $\mathbf{H}$ with a small number of dimensions and thus do not compute the sampling matrix $\mathbf{S}$ but instead use the full matrix. We first sampled a small number $s$ of rows from $\mathbf{A}^{(1)}$ and $\mathbf{A}^{(2)}$ and compared our bicriteria algorithm to the standard non-fair low-rank approximation algorithm baseline. We plot the minimum ratio for $s \in \{2, 3, \ldots, 21\}$, $k = 1$, and $p = 1$ over $10,000$ iterations for each setup in Figure 1a. Similarly, we plot both the minimum and average ratios for $s = 1000$, $k \in \{1, 2, \ldots, 7, 8\}$, and $p = 1$ over $200$ iterations in Figure 1b, where we permit the bicriteria solution to have rank $2k$. Finally, we compare the runtimes of the algorithms in Figure 1c, separating runtimes for our bicriteria into the total runtime `bicrit1` that includes the process of generating the Gaussian matrices and performing the Lewis weight sampling, as well as the runtime `bicrit2` for the step for extracting the factors

similar to SVD, which measures the runtime for extracting the factors. Across all experiments in Figure 1, we used Gaussian matrices $\mathbf{G}$ with 30 rows and $\mathbf{H}$ with 30 columns.

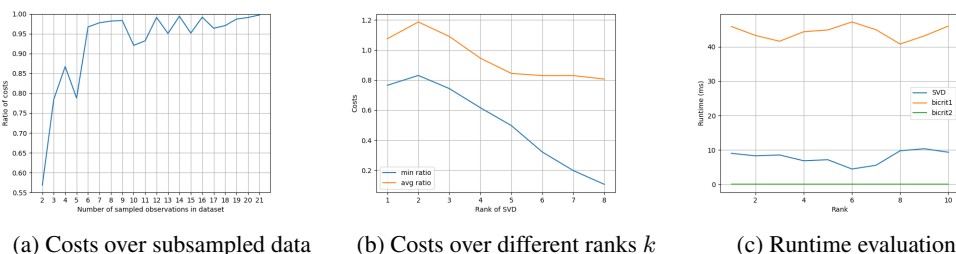

(a) Costs over subsampled data  (b) Costs over different ranks $k$  (c) Runtime evaluations

Fig. 1: Empirical evaluations on the Default Credit dataset.

**Results and discussion.** Our empirical evaluations serve as a simple proof-of-concept demonstrating that our bicriteria algorithm can perform significantly better for socially fair low-rank approximation, even without allowing for more than $k$ factors. In particular, all ratios in Figure 1a and most ratios in are less than 1, indicating that the cost of our algorithm is less than the cost of the baseline, so that smaller ratios represent better performance. In fact, in some cases the ratio was less than $0.6$, demonstrating the superiority of our algorithm. Finally, we show in Figure 1c that due to the small number of rows acquired by Lewis weight sampling, the final extraction of the factors is significantly faster by our algorithm. However, the main runtime bottleneck is the process of generating the Gaussian matrices and performing the subsequent Lewis weight sampling. As we use an iterative procedure to approximate the $L_p$ Lewis weights with 10 iterations, this translates to computing 10 matrix inversions. Hence it is likely this gap can be improved with further optimizations to Lewis weight approximation algorithms. We provide details on a number of additional empirical evaluations in Appendix D. In particular, we describe additional experiments on socially fair low-rank approximation on synthetic datasets and perform a quick investigation into socially fair linear regression, as additional motivation for studying fairness for randomized numerical linear algebra.

## 6 Conclusion

In this paper, we study algorithms for socially fair low-rank approximation and column subset selection. Although we show that even a constant-factor approximation to fair low-rank approximation requires exponential time under certain standard complexity hypotheses, we give an algorithm that is a substantial improvement over the naïve approach for any constant number of groups. We also give a bicriteria approximation algorithms for fair low-rank approximation and fair column subset selection that runs in polynomial time. Finally, we perform a number of empirical evaluations serving as a simple proof-of-concept demonstrating the practical nature of our theoretical findings. It is our hope that our work is an important step toward better understanding of fairness in numerical linear algebra.

Our work also leads to a number of interesting future directions. For example, there are various notions of fairness, including (1) *disparate impact*, which requires that the output of an algorithm respects a desired distribution that protects various subpopulations in the decision-making process [CKLV17, RS18, BIO$^+$19, BCFN19, BGK$^+$18, AEK$^+$20, EBTD20, DMV22, CXZC24], (2) *individual fairness*, where each element should incur a cost that is within a reasonable amount with respect to the overall dataset [JKL20, MV20, NC21, VY22], or (3) *representative fairness*, where the number of representative elements of each subpopulation should be relatively balanced [KAM19, JNN20, AKSZ22, NNJ22, TGOO22, HMV23]. A natural direction is the capabilities and limitations of randomized linear algebra for other notions of fairness. Another question is the efficient construction of $\varepsilon$-coresets with minimal size for socially fair subset selection. Finally, one can ask whether similar guarantees are possible in settings whether the input matrix is not centralized, but rather for example, distributed or arrives over a data stream. Indeed, there are known analogs [CDW18, BDM$^+$20, JLL$^+$21, WY23] for a number of the central tools used in the main algorithms of this paper.

## Acknowledgements

David P. Woodruff was supported in part by a Simons Investigator Award and NSF CCF-2335412. Samson Zhou is supported in part by NSF CCF-2335411. The work was conducted in part while Ali Vakilian, David P. Woodruff and Samson Zhou were visiting the Simons Institute for the Theory of Computing as part of the Sublinear Algorithms program.

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

# A Preliminaries

We use the notation $[n]$ to represent the set $\{1, \ldots, n\}$ for an integer $n \geq 1$. We use the notation $\text{poly}(n)$ to represent a fixed polynomial in $n$ and we use the notation $\text{polylog}(n)$ to represent $\text{poly}(\log n)$. We use $\text{poly}(n)$ to denote a fixed polynomial in $n$ and $\text{polylog}(n)$ to denote $\text{poly}(\log n)$. We say an event holds with high probability if it holds with probability $1 - \frac{1}{\text{poly}(n)}$.

We generally use bold-font variables to represent vectors and matrices, whereas we use default-font variables to represent scalars. For a matrix $\mathbf{A} \in \mathbb{R}^{n \times d}$, we use $\mathbf{A}_i$ to represent the $i$-th row of $\mathbf{A}$ and $\mathbf{A}^{(j)}$ to represent the $j$-th column of $\mathbf{A}$. We use $A_{i,j}$ to represent the entry in the $i$-th row and $j$-th column of $\mathbf{A}$. For $p \geq 1$, we use

$$\|\mathbf{A}\|_p = \left( \sum_{i \in [n]} \sum_{j \in [d]} A_{i,j}^p \right)^{1/p}$$

to represent the entrywise $L_p$ norm of $\mathbf{A}$ and we use

$$\|\mathbf{A}\|_F = \left( \sum_{i \in [n]} \sum_{j \in [d]} A_{i,j}^2 \right)^{1/2}$$

to represent the Frobenius norm of $\mathbf{A}$, which is simply the entrywise $L_2$ norm of $\mathbf{A}$. We use define the $L_{p,q}$ of $\mathbf{A}$ as the $L_p$ norm of the vector consisting of the $L_q$ norms of each row of $\mathbf{A}$, so that

$$\|\mathbf{A}\|_{p,q} = \left( \sum_{i \in [n]} \left( \sum_{j \in [d]} (A_{i,j})^q \right)^{p/q} \right)^{1/p}.$$

Similarly, we use $\|\mathbf{A}\|_{(p,q)}$ to denote the $L_p$ norm of the vector consisting of the $L_q$ norms of each column of $\mathbf{A}$. Equivalently, we have $\|\mathbf{A}\|_{(p,q)} = \|\mathbf{A}^\top\|_{p,q}$, so that

$$\|\mathbf{A}\|_{(p,q)} = \left( \sum_{j \in [d]} \left( \sum_{i \in [n]} (A_{i,j})^q \right)^{p/q} \right)^{1/p}.$$

We use $\circ$ to represent vertical stacking of matrices, so that

$$\mathbf{A}^{(1)} \circ \ldots \circ \mathbf{A}^{(m)} = \begin{bmatrix} \mathbf{A}^{(1)} \\ \vdots \\ \mathbf{A}^{(m)} \end{bmatrix}.$$

For a matrix $\mathbf{A} \in \mathbb{R}^{n \times d}$ of rank $k$, its singular decomposition is $\mathbf{A} = \mathbf{U}\mathbf{\Sigma}\mathbf{V}$, where $\mathbf{U} \in \mathbb{R}^{n \times k}$, $\mathbf{\Sigma} \in \mathbb{R}^{k \times k}$, $\mathbf{V} \in \mathbb{R}^{k \times d}$, so that the columns of $\mathbf{U}$ are orthonormal and the rows of $\mathbf{V}$ are orthonormal. The columns of $\mathbf{U}$ are called the left singular vectors of $\mathbf{A}$ and the rows of $\mathbf{V}$ are called the right singular vectors of $\mathbf{A}$.

## A.1 Regression and Low-Rank Approximation

In this section, we briefly describe some common techniques used to handle both regression and low-rank approximation, thus presenting multiple unified approaches for both problems. Thus in light of the abundance of techniques that can be used to handle both problems, it is somewhat surprising that socially fair regression and socially fair low-rank approximation exhibit vastly different complexities.

**Closed form solutions.** Given the regression problem $\min_{\mathbf{x} \in \mathbb{R}^d} \|\mathbf{A}\mathbf{x} - \mathbf{b}\|_2$ for an input matrix $\mathbf{A} \in \mathbb{R}^{n \times d}$ and a label vector $\mathbf{b} \in \mathbb{R}^n$, the closed form solution for the minimizer is $\mathbf{A}^\dagger \mathbf{b} = \text{argmin}_{\mathbf{x} \in \mathbb{R}^d} \|\mathbf{A}\mathbf{x} - \mathbf{b}\|_2$, where $\mathbf{A}^\dagger$ is the Moore-Penrose pseudoinverse of $\mathbf{A}$. Specifically, for a matrix $\mathbf{A} \in \mathbb{R}^{n \times d}$ written in its singular value decomposition $\mathbf{A} = \mathbf{U}\mathbf{\Sigma}\mathbf{v}$, we have $\mathbf{A}^\dagger =$

$\mathbf{V}^\top \mathbf{\Sigma}^{-1} \mathbf{U}^\top$. In particular, for if $\mathbf{A}$ has linearly independent rows, then $\mathbf{A}^\dagger = \mathbf{A}^\top (\mathbf{A}\mathbf{A}^\top)^{-1}$. On the other hand, if $\mathbf{A}$ has linearly independent columns, then $\mathbf{A}^\dagger = (\mathbf{A}^\top \mathbf{A})^{-1} \mathbf{A}^\top$.

Similarly, given an input matrix $\mathbf{A}$ and a rank parameter $k > 0$, there exists a closed form solution for the minimizer $\mathrm{argmin}_{\mathbf{V} \in \mathbb{R}^{k \times d}} \|\mathbf{A} - \mathbf{A}\mathbf{V}^\dagger \mathbf{V}\|_F^2$. Specifically, by the Eckart-Young-Mirsky theorem [EY36], the minimizer is the top $k$ right singular vectors of $\mathbf{A}$.

**Dimensionality reduction.** We next recall a unified set of dimensionality reduction techniques for both linear regression and low-rank approximation. We consider the "sketch-and-solve" paradigm, so that for both problems, we first acquire a low-dimension representation of the problem, and find the optimal solution in the low dimension using the above closed-form solutions. For "good" designs of the low-dimension representations, the low-dimension solution will also be near-optimal for the original problem.

We first observe that oblivious linear sketches serve as a common dimensionality reduction for both linear regression and low-rank approximation. For example, it is known [Woo14] that there exists a family of Gaussian random matrices $\mathcal{G}_1$ from which $\mathbf{S} \sim \mathcal{G}_1$ satisfies with high probability,

$$(1 - \varepsilon)\|\mathbf{S}\mathbf{A}\mathbf{x} - \mathbf{S}\mathbf{b}\|_2 \le \|\mathbf{A}\mathbf{x} - \mathbf{b}\|_2 \le (1 + \varepsilon)\|\mathbf{S}\mathbf{A}\mathbf{x} - \mathbf{S}\mathbf{b}\|_2,$$

simultaneously for all $\mathbf{x} \in \mathbb{R}^d$. Similarly, there exists [Woo14] a family of Gaussian random matrices $\mathcal{G}_2$ from which $\mathbf{S} \sim \mathcal{G}_1$ satisfies with high probability, that the row space of $\mathbf{S}\mathbf{A}$ contains a $(1 + \varepsilon)$-approximation of the optimal low-rank approximation to $\mathbf{A}$.

Alternatively, we can achieve dimensionality reduction for both linear regression and low-rank approximation by sampling a small subset of the input in related ways for both problems. For linear regression, we can generate a random matrix $\mathbf{S}$ by sampling rows of $[\mathbf{A} \quad \mathbf{b}]$ by their leverage scores [DMM06a, DMM06b, Mag10, Woo14]. In this manner, we again achieve a matrix $\mathbf{S}$ such that with high probability,

$$(1 - \varepsilon)\|\mathbf{S}\mathbf{A}\mathbf{x} - \mathbf{S}\mathbf{b}\|_2 \le \|\mathbf{A}\mathbf{x} - \mathbf{b}\|_2 \le (1 + \varepsilon)\|\mathbf{S}\mathbf{A}\mathbf{x} - \mathbf{S}\mathbf{b}\|_2,$$

simultaneously for all $\mathbf{x} \in \mathbb{R}^d$. For low-rank approximation, we can generate a random matrix $\mathbf{S}$ by sampling rows of $\mathbf{A}$ with the related ridge-leverage scores [CMM17]. Then with high probability, we have for all $\mathbf{V} \in \mathbb{R}^{k \times d}$,

$$(1 - \varepsilon)\|\mathbf{S}\mathbf{A} - \mathbf{S}\mathbf{A}\mathbf{V}^\dagger \mathbf{V}\|_F^2 \le \|\mathbf{A} - \mathbf{A}\mathbf{V}^\dagger \mathbf{V}\|_F^2 \le (1 + \varepsilon)\|\mathbf{S}\mathbf{A} - \mathbf{S}\mathbf{A}\mathbf{V}^\dagger \mathbf{V}\|_F^2.$$

# B   Missing Proofs from Section 3

## B.1   Lower Bound

We first show in Section B.1 that it is NP-hard to approximate fair low-rank approximation within any constant factor in polynomial time and moreover, under the exponential time hypothesis, it requires exponential time to achieve a constant factor approximation. We then give missing details from Section 3.1 and in Section 3.2.

Given points $\mathbf{v}^{(1)}, \ldots, \mathbf{v}^{(n)} \in \mathbb{R}^d$, their outer $(d - k)$-radius is defined as the minimum, over all $k$-dimensional linear subspaces, of the maximum Euclidean distance of these points to the subspace. We define this problem as $\mathsf{Subspace}(k, \infty)$. It is known that it is NP-hard to approximate the $\mathsf{Subspace}(n - 1, \infty)$ problem within any constant factor:

**Theorem B.1** ([BGK00, DTV11]). *The $\mathsf{Subspace}(n - 1, \infty)$ problem is NP-hard to approximate within any constant factor.*

Utilizing the NP-hardness of approximation of the $\mathsf{Subspace}(n - 1, \infty)$ problem, we show the NP-hardness of approximation of fair low-rank approximation.

**Theorem 1.1.** *Fair low-rank approximation is NP-hard to approximate within any constant factor.*

*Proof.* Given an instance $\mathbf{v}^{(1)}, \ldots, \mathbf{v}^{(n)} \in \mathbb{R}^d$ of $\mathsf{Subspace}(n - 1, \infty)$ with $n < d$, we set $\ell = k = n - 1$ and $\mathbf{A}^{(i)} = \mathbf{v}^{(i)}$ for all $i \in [n]$. Then for a $k$-dimensional linear subspace $\mathbf{V} \in \mathbb{R}^{k \times d}$, we have that $\|\mathbf{A}^{(i)}\mathbf{V}^\top \mathbf{V} - \mathbf{A}^{(i)}\|_F^2$ is the distance from $\mathbf{v}^{(i)}$ to the subspace. Hence, $\max_{i \in [\ell]} \|\mathbf{A}^{(i)}\mathbf{V}^\top \mathbf{V} -$

$\mathbf{A}^{(i)}\|_F^2$ is the maximum Euclidean distance of these points to the subspace and so the fair low-rank approximation problem is exactly $\mathsf{Subspace}(n-1,\infty)$. By Theorem B.1, the $\mathsf{Subspace}(n-1,\infty)$ problem is NP-hard to approximate within any constant factor. Thus, fair low-rank approximation is NP-hard to approximate within any constant factor. $\qquad\square$

We next introduce a standard complexity assumption beyond NP-hardness. Recall that in the 3-SAT problem, the input is a Boolean satisfiability problem written in conjunctive normal form, consisting of $n$ clauses, each with 3 literals, either a variable or the negation of a variable. The goal is to determine whether there exists a Boolean assignment to the variables to satisfy the formula.

**Hypothesis B.2** (Exponential time hypothesis [IP01]). *The 3-SAT problem requires $2^{\Omega(n)}$ runtime.*

Observe that while NP-hardness simply conjectures that the 3-SAT problem cannot be solved in polynomial time, the exponential time hypothesis conjectures that the 3-SAT problem requires *exponential* time.

We remark that in the context of Theorem B.1, [BGK00] showed the hardness of approximation of $\mathsf{Subspace}(n-1,\infty)$ through a reduction from the Max-Not-All-Equal-3-SAT problem, whose NP-hardness itself is shown through a reduction from 3-SAT. Thus under the exponential time hypothesis, Max-Not-All-Equal-3-SAT problem requires $2^{\Omega(n)}$ to solve. Then it follows that:

**Theorem B.3** ([BGK00, DTV11]). *Assuming the exponential time hypothesis, then the $\mathsf{Subspace}(n-1,\infty)$ problem requires $2^{n^{\Omega(1)}}$ time to approximate within any constant factor.*

It follows that under the exponential time hypothesis, any constant-factor approximation to socially fair low-rank approximation requires exponential time.

**Theorem 1.2.** *Under the exponential time hypothesis, the fair low-rank approximation requires $2^{k^{\Omega(1)}}$ time to approximate within any constant factor.*

*Proof.* Given an instance $\mathbf{v}^{(1)},\ldots,\mathbf{v}^{(k)}\in\mathbb{R}^d$ of $\mathsf{Subspace}(k-1,\infty)$ with $k<d$, we set $\ell=k-1$ and $\mathbf{A}^{(i)}=\mathbf{v}^{(i)}$ for all $i\in[k]$. Then for a $(k-1)$-dimensional linear subspace $\mathbf{V}\in\mathbb{R}^{(k-1)\times d}$, we have that $\|\mathbf{A}^{(i)}\mathbf{V}^\top\mathbf{V}-\mathbf{A}^{(i)}\|_F^2$ is the distance from $\mathbf{v}^{(i)}$ to the subspace. Hence, $\max_{i\in[\ell]}\|\mathbf{A}^{(i)}\mathbf{V}^\top\mathbf{V}-\mathbf{A}^{(i)}\|_F^2$ is the maximum Euclidean distance of these points to the subspace and so the fair low-rank approximation problem is exactly $\mathsf{Subspace}(k-1,\infty)$. By Theorem B.3, the $\mathsf{Subspace}(k-1,\infty)$ problem requires $2^{k^{\Omega(1)}}$ time to approximate within any constant factor. Thus, fair low-rank approximation requires $2^{k^{\Omega(1)}}$ time to approximate within any constant factor. $\qquad\square$

### B.2 Missing Proofs from Section 3.1

We first recall the following result for polynomial system satisfiability solvers.

**Theorem B.4** ([Ren92a, Ren92b, BPR96]). *Given a polynomial system $P(x_1,\ldots,x_n)$ over real numbers and $m$ polynomial constraints $f_i(x_1,\ldots,x_n)\otimes_i 0$, where $\otimes\in\{>,\geq,=,\neq,\leq,<\}$ for all $i\in[m]$, let $d$ denote the maximum degree of all the polynomial constraints and let $B$ denote the maximum size of the bit representation of the coefficients of all the polynomial constraints. Then there exists an algorithm that determines whether there exists a solution to the polynomial system $P$ in time $(md)^{\mathcal{O}(n)}\cdot\mathrm{poly}(B)$.*

To apply Theorem B.4, we utilize the following statement upper bounding the sizes of the bit representation of the coefficients of the polynomial constraints in our system.

**Theorem B.5** ([JPT13]). *Let $\mathcal{T}=\{x\in\mathbb{R}^n\mid f_1(x)\geq 0,\ldots,f_m(x)\geq 0\}$ be defined by $m$ polynomials $f_i(x_1,\ldots,x_n)$ for $i\in[m]$ with degrees bounded by an even integer $d$ and coefficients of magnitude at most $M$. Let $\mathcal{C}$ be a compact connected component of $\mathcal{T}$. Let $g(x_1,\ldots,x_n)$ be a polynomial of degree at most $d$ with integer coefficients of magnitude at most $M$. Then the minimum nonzero magnitude that $g$ takes over $\mathcal{C}$ is at least $(2^{4-n/2}\widetilde{M}d^n)^{-n2^n d^n}$, where $\widetilde{M}=\max(M,2n+2m)$.*

To perform dimensionality reduction, recall the following definition of affine embedding.

**Definition B.6** (Affine embedding). *We say a matrix $\mathbf{S} \in \mathbb{R}^{n \times m}$ is an affine embedding for a matrix $\mathbf{A} \in \mathbb{R}^{d \times n}$ and a vector $\mathbf{b} \in \mathbb{R}^n$ if we have*

$$(1 - \varepsilon)\|\mathbf{x}\mathbf{A} - \mathbf{b}\|_F^2 \leq \|\mathbf{x}\mathbf{A}\mathbf{S} - \mathbf{b}\mathbf{S}\|_F^2 \leq (1 + \varepsilon)\|\mathbf{x}\mathbf{A} - \mathbf{b}\|_F^2,$$

*for all vectors $\mathbf{x} \in \mathbb{R}^d$.*

We then apply the following affine embedding construction.

**Lemma B.7** (Lemma 11 in [CEM$^+$15]). *Given $\delta, \varepsilon \in (0, 1)$ and a rank parameter $k > 0$, let $m = \mathcal{O}\left(\frac{k^2}{\varepsilon^2} \log \frac{1}{\delta}\right)$. For any matrix $\mathbf{A} \in \mathbb{R}^{d \times n}$, there exists a family $\mathcal{S}$ of random matrices in $\mathbb{R}^{n \times m}$, such that for $\mathbf{S} \sim \mathcal{S}$, we have that with probability at least $1 - \delta$, $\mathbf{S}$ is a one-sided affine embedding for a matrix $\mathbf{A} \in \mathbb{R}^{d \times n}$ and a vector $\mathbf{b} \in \mathbb{R}^n$.*

We now show a crucial structural property that allows us to distinguish between the case where a guess $\alpha$ for the optimal value OPT exceeds $(1 + \varepsilon)$OPT or is smaller than $(1 - \varepsilon)$OPT by simply looking at a polynomial system solver on an affine embedding.

**Lemma 3.1.** *Let $\mathbf{V} \in \mathbb{R}^{k \times d}$ be the optimal solution to the fair low-rank approximation problem for inputs $\mathbf{A}^{(1)}, \ldots, \mathbf{A}^{(\ell)}$, where $\mathbf{A}^{(i)} \in \mathbb{R}^{n_i \times d}$, and suppose $\mathsf{OPT} = \max_{i \in [\ell]} \|\mathbf{A}^{(i)} \mathbf{V}^\dagger \mathbf{V} - \mathbf{A}^{(i)}\|_F^2$. Let $\mathbf{S}$ be an affine embedding for $\mathbf{V}$ and let $\mathbf{W} = (\mathbf{V}\mathbf{S})^\dagger \in \mathbb{R}^{k \times m}$. For $i \in [\ell]$, let $\mathbf{Z}^{(i)} = \mathbf{A}^{(i)} \mathbf{S} \mathbf{W} \in \mathbb{R}^{n_i \times k}$ and $\mathbf{R}^{(i)} \in \mathbb{R}^{k \times k}$ be defined so that $\mathbf{A}^{(i)} \mathbf{S} \mathbf{W} \mathbf{R}^{(i)}$ has orthonormal columns. If $\alpha \geq (1 + \varepsilon) \cdot \mathsf{OPT}$, then for each $i \in [\ell]$, $\alpha \geq \|(\mathbf{A}^{(i)} \mathbf{S} \mathbf{W} \mathbf{R}^{(i)})(\mathbf{A}^{(i)} \mathbf{S} \mathbf{W} \mathbf{R}^{(i)})^\dagger \mathbf{A}^{(i)} - \mathbf{A}^{(i)}\|_F^2$. If $\alpha < (1 - \varepsilon) \cdot \mathsf{OPT}$, then there exists $i \in [\ell]$, such that $\alpha < \|(\mathbf{A}^{(i)} \mathbf{S} \mathbf{W} \mathbf{R}^{(i)})(\mathbf{A}^{(i)} \mathbf{S} \mathbf{W} \mathbf{R}^{(i)})^\dagger \mathbf{A}^{(i)} - \mathbf{A}^{(i)}\|_F^2$.*

*Proof.* By the Pythagorean theorem, we have that

$$(\mathbf{A}^{(i)} \mathbf{S})(\mathbf{V}\mathbf{S})^\dagger = \underset{\mathbf{X}^{(i)}}{\operatorname{argmin}} \|\mathbf{X}^{(i)} \mathbf{V}\mathbf{S} - \mathbf{A}^{(i)} \mathbf{S}\|_F^2.$$

Thus, $\mathbf{Z}^{(i)} = \operatorname{argmin}_{\mathbf{X}^{(i)}} \|\mathbf{X}^{(i)} \mathbf{V}\mathbf{S} - \mathbf{A}^{(i)} \mathbf{S}\|_F^2$ for $\mathbf{Z}^{(i)} = \mathbf{A}^{(i)} \mathbf{S} \mathbf{W} \in \mathbb{R}^{n_i \times k}$ and $\mathbf{W} = (\mathbf{V}\mathbf{S})^\dagger \in \mathbb{R}^{m \times k}$.

Let $\mathbf{R}^{(i)} \in \mathbb{R}^{k \times k}$ be defined so that $\mathbf{A}^{(i)} \mathbf{S} \mathbf{W} \mathbf{R}^{(i)}$ has orthonormal columns. Thus, we have

$$\begin{aligned}
\|(\mathbf{Z}^{(i)} \mathbf{R}^{(i)})(\mathbf{Z}^{(i)} \mathbf{R}^{(i)})^\dagger \mathbf{A}^{(i)} \mathbf{S} - \mathbf{A}^{(i)} \mathbf{S}\|_F^2 &= \|(\mathbf{Z}^{(i)})(\mathbf{Z}^{(i)})^\dagger \mathbf{A}^{(i)} \mathbf{S} - \mathbf{A}^{(i)} \mathbf{S}\|_F^2 \\
&= \|\mathbf{Z}^{(i)} \mathbf{V}\mathbf{S} - \mathbf{A}^{(i)} \mathbf{S}\|_F^2 \\
&= \min_{\mathbf{X}^{(i)}} \|\mathbf{X}^{(i)} \mathbf{V}\mathbf{S} - \mathbf{A}^{(i)} \mathbf{S}\|_F^2,
\end{aligned}$$

by the definition of $\mathbf{Z}^{(i)}$.

Suppose $\alpha \geq (1 + \varepsilon) \cdot \mathsf{OPT}$. Since $\mathbf{S}$ is an affine embedding for $\mathbf{V}$, then we have that for all $\mathbf{X}^{(i)} \in \mathbb{R}^{n_i \times k}$,

$$\|\mathbf{X}^{(i)} \mathbf{V}\mathbf{S} - \mathbf{A}^{(i)} \mathbf{S}\|_F^2 \leq (1 + \varepsilon)\|\mathbf{X}^{(i)} \mathbf{V} - \mathbf{A}^{(i)}\|_F^2.$$

In particular, we have

$$\min_{\mathbf{X}^{(i)}} \|\mathbf{X}^{(i)} \mathbf{V}\mathbf{S} - \mathbf{A}^{(i)} \mathbf{S}\|_F^2 \leq (1 + \varepsilon) \min_{\mathbf{X}^{(i)}} \|\mathbf{X}^{(i)} \mathbf{V} - \mathbf{A}^{(i)}\|_F^2 \leq (1 + \varepsilon)\mathsf{OPT}.$$

Then from the above argument, we have

$$\begin{aligned}
\|(\mathbf{Z}^{(i)} \mathbf{R}^{(i)})(\mathbf{Z}^{(i)} \mathbf{R}^{(i)})^\dagger \mathbf{A}^{(i)} \mathbf{S} - \mathbf{A}^{(i)} \mathbf{S}\|_F^2 &= \min_{\mathbf{X}^{(i)}} \|\mathbf{X}^{(i)} \mathbf{V}\mathbf{S} - \mathbf{A}^{(i)} \mathbf{S}\|_F^2 \\
&\leq (1 + \varepsilon)\mathsf{OPT} \leq \alpha.
\end{aligned}$$

Since $\mathbf{Z}^{(i)} = \mathbf{A}^{(i)} \mathbf{S} \mathbf{W} \in \mathbb{R}^{n_i \times k}$, then for $\alpha \geq (1 + \varepsilon) \cdot \mathsf{OPT}$, we have that for each $i \in [\ell]$,

$$\alpha \geq \|(\mathbf{A}^{(i)} \mathbf{S} \mathbf{W} \mathbf{R}^{(i)})(\mathbf{A}^{(i)} \mathbf{S} \mathbf{W} \mathbf{R}^{(i)})^\dagger \mathbf{A}^{(i)} - \mathbf{A}^{(i)}\|_F^2.$$

On the other hand, suppose $\alpha < (1 - \varepsilon) \cdot \mathsf{OPT}$. Let $i \in [\ell]$ be fixed so that

$$\mathsf{OPT} = \min_{\mathbf{X}^{(i)}} \|\mathbf{X}^{(i)} \mathbf{V} - \mathbf{A}^{(i)}\|_F^2.$$

Since $\mathbf{S}$ is an affine embedding for $\mathbf{V}$, we have that for all $\mathbf{X}^{(i)} \in \mathbb{R}^{n_i \times k}$,

$$(1 - \varepsilon)\|\mathbf{X}^{(i)}\mathbf{V} - \mathbf{A}^{(i)}\|_F^2 \leq \|\mathbf{X}^{(i)}\mathbf{V}\mathbf{S} - \mathbf{A}^{(i)}\mathbf{S}\|_F^2,$$

Therefore,

$$(1 - \varepsilon)\min_{\mathbf{X}^{(i)}}\|\mathbf{X}^{(i)}\mathbf{V} - \mathbf{A}^{(i)}\|_F^2 \leq \min_{\mathbf{X}^{(i)}}\|\mathbf{X}^{(i)}\mathbf{V}\mathbf{S} - \mathbf{A}^{(i)}\mathbf{S}\|_F^2$$

From the above, we have

$$\|(\mathbf{Z}^{(i)}\mathbf{R}^{(i)})(\mathbf{Z}^{(i)}\mathbf{R}^{(i)})^\dagger\mathbf{A}^{(i)}\mathbf{S} - \mathbf{A}^{(i)}\mathbf{S}\|_F^2 = \min_{\mathbf{X}^{(i)}}\|\mathbf{X}^{(i)}\mathbf{V}\mathbf{S} - \mathbf{A}^{(i)}\mathbf{S}\|_F^2.$$

Hence, putting these relations together,

$$\begin{aligned}
\alpha < (1 - \varepsilon)\mathsf{OPT} &= (1 - \varepsilon)\min_{\mathbf{X}^{(i)}}\|\mathbf{X}^{(i)}\mathbf{V} - \mathbf{A}^{(i)}\|_F^2 \\
&\leq \min_{\mathbf{X}^{(i)}}\|\mathbf{X}^{(i)}\mathbf{V}\mathbf{S} - \mathbf{A}^{(i)}\mathbf{S}\|_F^2 \\
&= \|(\mathbf{Z}^{(i)}\mathbf{R}^{(i)})(\mathbf{Z}^{(i)}\mathbf{R}^{(i)})^\dagger\mathbf{A}^{(i)}\mathbf{S} - \mathbf{A}^{(i)}\mathbf{S}\|_F^2,
\end{aligned}$$

as desired. $\qquad\square$

We can thus utilize the structural property of Lemma 3.1 by using the polynomial system solver in Algorithm 1 on an affine embedding.

**Corollary B.8.** *If $\alpha \geq (1 + \varepsilon) \cdot \mathsf{OPT}$, then Algorithm 1 outputs a vector $\mathbf{U} \in \mathbb{R}^{n \times k}$ such that*

$$\alpha \geq \|\mathbf{U}\mathbf{U}^\dagger\mathbf{A}^{(i)} - \mathbf{A}^{(i)}\|_F^2.$$

*If $\alpha < (1 - \varepsilon) \cdot \mathsf{OPT}$, then Algorithm 1 outputs $\perp$*

Correctness of Algorithm 2 then follows from Corollary B.8 and binary search on $\alpha$.

We now analyze the runtime of Algorithm 2.

**Lemma B.9.** *The runtime of Algorithm 2 is at most $\frac{1}{\varepsilon}\operatorname{poly}(n) \cdot (2\ell)^{\mathcal{O}(N)}$, for $n = \sum_{i=1}^{\ell} n_i$ and $N = \operatorname{poly}\left(\ell, k, \frac{1}{\varepsilon}\right)$.*

*Proof.* Suppose the coefficients of $\mathbf{A}^{(i)}$ are bounded in magnitude by $2^{\operatorname{poly}(n)}$, where $n = \sum_{i=1}^{\ell} n_i$. The number of variables in the polynomial system is at most

$$N := 2mk + \ell k^2 = \operatorname{poly}\left(\ell, k, \frac{1}{\varepsilon}\right).$$

Each of the $\mathcal{O}(\ell)$ polynomial constraints has degree at most 20. Thus by Theorem B.5, the minimum nonzero magnitude that any polynomial constraint takes over $\mathcal{C}$ is at least $(2^{4-N/2}2^{\operatorname{poly}(n)}20^N)^{-N2^N20^N}$. Hence, the maximum bit representation required is $B = \operatorname{poly}(n) \cdot 2^{\mathcal{O}(N)}$. By Theorem B.4, the runtime of the polynomial system solver is at most $(\mathcal{O}(\ell) \cdot 20)^{\mathcal{O}(N)} \cdot \operatorname{poly}(B) = \operatorname{poly}(n) \cdot (2\ell)^{\mathcal{O}(N)}$. We require at most $\mathcal{O}\left(\frac{1}{\varepsilon}\log\ell\right)$ iterations of the polynomial system solver. Thus, the total runtime is at most $\frac{1}{\varepsilon}\operatorname{poly}(n) \cdot (2\ell)^{\mathcal{O}(N)}$. $\qquad\square$

Putting things together, we have:

**Theorem 1.3.** *Given an accuracy parameter $\varepsilon \in (0, 1)$, there exists an algorithm which outputs $\widehat{\mathbf{V}} \in \mathbb{R}^{k \times d}$ such that with probability at least $\frac{2}{3}$, $\max_{i \in [\ell]}\|\mathbf{A}^{(i)}(\widehat{\mathbf{V}})^\dagger\widehat{\mathbf{V}} - \mathbf{A}^{(i)}\|_F \leq (1 + \varepsilon) \cdot \min_{\mathbf{V} \in \mathbb{R}^{k \times d}}\max_{i \in [\ell]}\|\mathbf{A}^{(i)}\mathbf{V}^\dagger\mathbf{V} - \mathbf{A}^{(i)}\|_F$. The algorithm uses runtime $\frac{1}{\varepsilon}\operatorname{poly}(n) \cdot (2\ell)^{\mathcal{O}(N)}$, for $n = \sum_{i=1}^{\ell} n_i$ and $N = \operatorname{poly}\left(\ell, k, \frac{1}{\varepsilon}\right)$.*

## B.3 Missing Proofs from Section 3.2

**Lemma B.10.** *Let $\varepsilon \in (0,1)$ and $\mathbf{x} \in \mathbb{R}^{\ell}$ and let $p = \mathcal{O}\left(\frac{1}{\varepsilon}\log \ell\right)$. Then $\|\mathbf{x}\|_{\infty} \leq \|\mathbf{x}\|_p \leq (1+\varepsilon)\|\mathbf{x}\|_{\infty}$.*

*Proof.* Since it is clear that $\|\mathbf{x}\|_{\infty} \leq \|\mathbf{x}\|_p$, then it remains to prove $\|\mathbf{x}\|_p \leq (1+\varepsilon)\|\mathbf{x}\|_{\infty}$. Note that we have $\|\mathbf{x}\|_p^p \leq \|\mathbf{x}\|_{\infty}^p \cdot \ell$. To achieve $\ell^{1/p} \leq (1+\varepsilon)$, it suffices to have $\frac{1}{p}\log \ell \leq \log(1+\varepsilon)$. Since $\log(1+\varepsilon) = \mathcal{O}(\varepsilon)$ for $\varepsilon \in (0,1)$, then for $p = \mathcal{O}\left(\frac{1}{\varepsilon}\log \ell\right)$, we have $\ell^{1/p} \leq (1+\varepsilon)$, and the desired claim follows. $\qquad\square$

**Lemma 3.5.** *Let $\widetilde{\mathbf{V}}$ be the output of Algorithm 3. Then with probability at least $\frac{9}{10}$, $\max_{i \in [\ell]} \|\mathbf{A}^{(i)}(\widetilde{\mathbf{V}})^{\dagger}\widetilde{\mathbf{V}} - \mathbf{A}^{(i)}\|_F$ is at most $\ell^c \cdot 2^{1/c} \cdot \mathcal{O}\left(k(\log\log k)(\log d)\right)\max_{i\in[\ell]}\|\mathbf{A}^{(i)}(\widetilde{\mathbf{V}})^{\dagger}\widetilde{\mathbf{V}} - \mathbf{A}^{(i)}\|_F$, where $c$ is the trade-off parameter input.*

*Proof.* We have

$$\max_{i\in[\ell]}\|\mathbf{A}^{(i)}(\widetilde{\mathbf{V}})^{\dagger}\widetilde{\mathbf{V}} - \mathbf{A}^{(i)}\|_F \leq \left(\sum_{i\in[\ell]}\|\mathbf{A}^{(i)}(\widetilde{\mathbf{V}})^{\dagger}\widetilde{\mathbf{V}} - \mathbf{A}^{(i)}\|_F^p\right)^{1/p}.$$

Let $\mathbf{A} = \mathbf{A}^{(1)} \circ \ldots \circ \mathbf{A}^{(\ell)}$ and let

$$\widetilde{\mathbf{U}} := \mathbf{AHS}.$$

For $i \in [\ell]$, let $\widetilde{\mathbf{U}^{(i)}}$ be the matrix of $\widetilde{\mathbf{U}}$ whose rows correspond with the rows of $\mathbf{A}^{(i)}$ in $\mathbf{A}$, i.e., let $\widetilde{\mathbf{U}^{(i)}}$ be the $i$-th block of rows of $\widetilde{\mathbf{U}}$.

By the optimality of $\mathbf{A}^{(i)}(\widetilde{\mathbf{V}})^{\dagger}$ with respect to the Frobenius norm, we have

$$\left(\sum_{i\in[\ell]}\|\mathbf{A}^{(i)}(\widetilde{\mathbf{V}})^{\dagger}\widetilde{\mathbf{V}} - \mathbf{A}^{(i)}\|_F^p\right)^{1/p} \leq \left(\sum_{i\in[\ell]}\|\widetilde{\mathbf{U}^{(i)}}\widetilde{\mathbf{V}} - \mathbf{A}^{(i)}\|_F^p\right)^{1/p}$$

By Dvoretzky's Theorem, Theorem 3.2, with distortion $\varepsilon = \Theta(1)$, we have that with probability at least $0.99$,

$$\left(\sum_{i\in[\ell]}\|\widetilde{\mathbf{U}^{(i)}}\widetilde{\mathbf{V}} - \mathbf{A}^{(i)}\|_F^p\right)^{1/p} \leq 2\left(\sum_{i\in[\ell]}\|\mathbf{G}\widetilde{\mathbf{U}^{(i)}}\widetilde{\mathbf{V}}\mathbf{H} - \mathbf{G}\mathbf{A}^{(i)}\mathbf{H}\|_p^p\right)^{1/p},$$

where we use $\|\cdot\|_p$ to denote the entry-wise $p$-norm. Writing $\widetilde{\mathbf{U}} = \widetilde{\mathbf{U}^{(1)}} \circ \ldots \circ \widetilde{\mathbf{U}^{(\ell)}}$, then we have

$$\left(\sum_{i\in[\ell]}\|\mathbf{G}\widetilde{\mathbf{U}^{(i)}}\widetilde{\mathbf{V}}\mathbf{H} - \mathbf{G}\mathbf{A}^{(i)}\mathbf{H}\|_p^p\right)^{1/p} = \|\mathbf{G}\widetilde{\mathbf{U}}\widetilde{\mathbf{V}}\mathbf{H} - \mathbf{G}\mathbf{A}\mathbf{H}\|_p.$$

By the choice of the Lewis weight sampling matrix $\mathbf{T}$, we have that with probability $0.99$,

$$\|\mathbf{G}\widetilde{\mathbf{U}}\widetilde{\mathbf{V}}\mathbf{H} - \mathbf{G}\mathbf{A}\mathbf{H}\|_p \leq 2\|\mathbf{T}\mathbf{G}\widetilde{\mathbf{U}}\widetilde{\mathbf{V}}\mathbf{H} - \mathbf{T}\mathbf{G}\mathbf{A}\mathbf{H}\|_p$$

$$\leq 2\|\mathbf{T}\mathbf{G}\widetilde{\mathbf{U}}\widetilde{\mathbf{V}}\mathbf{H} - \mathbf{T}\mathbf{G}\mathbf{A}\mathbf{H}\|_{(p,2)}$$

$$= 2\|\mathbf{T}\mathbf{G}\mathbf{A}\mathbf{H}\mathbf{S}(\mathbf{T}\mathbf{G}\mathbf{A}\mathbf{H}\mathbf{S})^{\dagger}(\mathbf{T}\mathbf{G}\mathbf{A})\mathbf{H} - \mathbf{T}\mathbf{G}\mathbf{A}\mathbf{H}\|_{(p,2)},$$

where $\|\mathbf{M}\|_{(p,2)}$ denotes the $L_p$ norm of the vector consisting of the $L_2$ norms of the columns of $\mathbf{M}$, and the last equality follows due to our setting of $\widetilde{\mathbf{U}} = \mathbf{AHS}$ and $\widetilde{\mathbf{V}} \leftarrow (\mathbf{TGAHS})^{\dagger}(\mathbf{TGA})$, the latter in Algorithm 3. By optimality of $(\mathbf{TGAHS})^{\dagger}(\mathbf{TGA})\mathbf{H}$ for the choice of $\mathbf{X}$ in the minimization problem

$$\min_{\mathbf{X}\in\mathbb{R}^{t\times d'}} \|\mathbf{TGAHSX} - \mathbf{TGAH}\|_{(p,2)},$$

we have

$$\|\mathbf{TGAHS}(\mathbf{TGAHS})^\dagger(\mathbf{TGA})\mathbf{H} - \mathbf{TGAH}\|_{(p,2)} = \min_{\mathbf{X} \in \mathbb{R}^{t \times d'}} \|\mathbf{TGAHSX} - \mathbf{TGAH}\|_{(p,2)}.$$

Since $\mathbf{S} \in \mathbb{R}^{n' \times t}$ and $\mathbf{T}$ is a Lewis weight sampling matrix for $\mathbf{GAHSX} - \mathbf{GAH}$, then $\mathbf{T}$ has $t$ rows, where $t = \mathcal{O}\left(k(\log\log k)(\log^2 d)\right)$ by Theorem 3.3. Thus, each column of $\mathbf{TGAH}$ has $t$ entries, so that

$$\min_{\mathbf{X} \in \mathbb{R}^{t \times d'}} \|\mathbf{TGAHSX} - \mathbf{TGAH}\|_{(p,2)} \leq \sqrt{t} \min_{\mathbf{X} \in \mathbb{R}^{t \times d'}} \|\mathbf{TGAHSX} - \mathbf{TGAH}\|_p.$$

By the choice of the Lewis weight sampling matrix $\mathbf{T}$, we have that with probability 0.99,

$$\min_{\mathbf{X} \in \mathbb{R}^{t \times d'}} \|\mathbf{TGAHSX} - \mathbf{TGAH}\|_p \leq 2 \min_{\mathbf{X} \in \mathbb{R}^{t \times d'}} \|\mathbf{GAHSX} - \mathbf{GAH}\|_p.$$

By Theorem 3.3, we have that with probability 0.99,

$$\min_{\mathbf{X} \in \mathbb{R}^{t \times d'}} \|\mathbf{GAHSX} - \mathbf{GAH}\|_p \leq 2^p \cdot \mathcal{O}\left(\sqrt{s}\right) \cdot \min_{\mathbf{U} \in \mathbb{R}^{n' \times t}, \mathbf{V} \in \mathbb{R}^{t \times d'}} \|\mathbf{UV} - \mathbf{GAH}\|_p,$$

for $s = \mathcal{O}\left(k \log\log k\right)$. Let $\mathbf{V}^* = \operatorname{argmin}_{\mathbf{V} \in \mathbb{R}^{k \times d}} \max_{i \in [\ell]} \|\mathbf{A}^{(i)} - \mathbf{A}^{(i)} \mathbf{V}^\dagger \mathbf{V}\|_p$. Then since $\mathbf{UV}$ has rank $t$ with $t \geq k$, we have

$$\min_{\mathbf{U} \in \mathbb{R}^{n' \times t}, \mathbf{V} \in \mathbb{R}^{t \times d'}} \|\mathbf{UV} - \mathbf{GAH}\|_p \leq \|\mathbf{GA}(\mathbf{V}^*)^\dagger \mathbf{V}^* \mathbf{H} - \mathbf{GAH}\|_p$$

$$= \left( \sum_{i \in [\ell]} \|\mathbf{GA}^{(i)}(\mathbf{V}^*)^\dagger \mathbf{V}^* \mathbf{H} - \mathbf{GA}^{(i)} \mathbf{H}\|_p^p \right)^{1/p}.$$

By Dvoretzky's Theorem, Theorem 3.2, with distortion $\varepsilon = \Theta(1)$, we have that with probability at least 0.99,

$$\left( \sum_{i \in [\ell]} \|\mathbf{GA}^{(i)}(\mathbf{V}^*)^\dagger \mathbf{V}^* \mathbf{H} - \mathbf{GA}^{(i)} \mathbf{H}\|_p^p \right)^{1/p} \leq 2 \left( \sum_{i \in [\ell]} \|\mathbf{A}^{(i)}(\mathbf{V}^*)^\dagger \mathbf{V}^* - \mathbf{A}^{(i)}\|_F^p \right)^{1/p}.$$

For $p = c \log \ell$ with $c < 1$, we have

$$\left( \sum_{i \in [\ell]} \|\mathbf{A}^{(i)}(\mathbf{V}^*)^\dagger \mathbf{V}^* - \mathbf{A}^{(i)}\|_F^p \right)^{1/p} \leq 2^{1/c} \max_{i \in [\ell]} \|\mathbf{A}^{(i)}(\widetilde{\mathbf{V}})^\dagger \widetilde{\mathbf{V}} - \mathbf{A}^{(i)}\|_F.$$

Putting together these inequalities successively, we ultimately have

$$\max_{i \in [\ell]} \|\mathbf{A}^{(i)}(\widetilde{\mathbf{V}})^\dagger \widetilde{\mathbf{V}} - \mathbf{A}^{(i)}\|_F \leq 2^p \cdot 2^{1/c} \cdot \mathcal{O}\left(\sqrt{st}\right) \max_{i \in [\ell]} \|\mathbf{A}^{(i)}(\widetilde{\mathbf{V}})^\dagger \widetilde{\mathbf{V}} - \mathbf{A}^{(i)}\|_F,$$

for $p = c \log \ell$, $s = \mathcal{O}\left(k \log\log k\right)$, and $t = \mathcal{O}\left(k(\log\log k)(\log^2 d)\right)$. Therefore, we have

$$\max_{i \in [\ell]} \|\mathbf{A}^{(i)}(\widetilde{\mathbf{V}})^\dagger \widetilde{\mathbf{V}} - \mathbf{A}^{(i)}\|_F \leq \ell^c \cdot 2^{1/c} \cdot \mathcal{O}\left(k(\log\log k)(\log d)\right) \max_{i \in [\ell]} \|\mathbf{A}^{(i)}(\widetilde{\mathbf{V}})^\dagger \widetilde{\mathbf{V}} - \mathbf{A}^{(i)}\|_F.$$

$\square$

## C   Missing Proofs from Section 4

It is known that the sum of the leverage scores of the rows of a matrix can be bounded by the rank of the matrix.

**Theorem C.1** (Generalization of Foster's Theorem, [Fos53]). *Given a matrix $\mathbf{M} \in \mathbb{R}^{n \times d}$, the sum of its leverage scores is* $\operatorname{rank}(\mathbf{M})$.

By sampling rows proportional to their leverage scores, we can obtain a subspace embedding as follows:

**Theorem C.2** (Leverage score sampling). *[DMM06a, DMM06b, Mag10, Woo14] Given a matrix* $\mathbf{M} \in \mathbb{R}^{n \times d}$, *let* $\sigma_i$ *be the leverage score of the $i$-th row of* $\mathbf{M}$. *Suppose* $p_i = \min(1, \sigma_i \log n)$ *for each* $i \in [n]$ *and let* $\mathbf{S}$ *be a random diagonal matrix so that the $i$-th diagonal entry of* $\mathbf{S}$ *is* $\frac{1}{\sqrt{p_i}}$ *with probability* $p_i$ *and* $0$ *with probability* $1 - p_i$. *Then for all vectors* $\mathbf{v} \in \mathbb{R}^n$,

$$\mathbb{E}\left[\|\mathbf{S}\mathbf{v}\|_2^2\right] = \|\mathbf{v}\|_2^2$$

*and with probability at least* $0.99$, *for all vectors* $\mathbf{x} \in \mathbb{R}^d$

$$\frac{99}{100}\|\mathbf{M}\mathbf{x}\|_2 \leq \|\mathbf{S}\mathbf{M}\mathbf{x}\|_2 \leq \frac{101}{100}\|\mathbf{M}\mathbf{x}\|_2.$$

*Moreover,* $\mathbf{S}$ *has at most* $\mathcal{O}(d \log n)$ *nonzero entries with high probability.*

**Lemma 4.2.** *Given a matrix* $\mathbf{M} \in \mathbb{R}^{n \times d}$, *let* $\mathbf{S}$ *be a matrix such that for all* $\mathbf{x} \in \mathbb{R}^d$ *and* $\mathbf{v} \in \mathbb{R}^n$,

$$\frac{11}{12}\|\mathbf{M}\mathbf{x}\|_2 \leq \|\mathbf{S}\mathbf{M}\mathbf{x}\|_2 \leq \frac{13}{12}\|\mathbf{M}\mathbf{x}\|_2, \ \mathbb{E}\left[\|\mathbf{S}\mathbf{v}\|_2^2\right] = \|\mathbf{v}\|_2^2.$$

*For a fixed* $\mathbf{B} \in \mathbb{R}^{n \times m}$ *where* $\mathbf{B} = \mathbf{b}_1 \circ \ldots \circ \mathbf{b}_m$ *with* $\mathbf{b}_i \in \mathbb{R}^n$ *for* $i \in [m]$, *let* $\widetilde{\mathbf{x}}_i = (\mathbf{S}\mathbf{M})^\dagger (\mathbf{S}\mathbf{b}_i)$. *Let* $\widetilde{\mathbf{X}} = \widetilde{\mathbf{x}_1} \circ \ldots \circ \widetilde{\mathbf{x}_m}$. *Then with probability at least* $0.97$, $\|\mathbf{M}\widetilde{\mathbf{X}} - \mathbf{B}\|_2 \leq 99 \min_{\mathbf{X}} \|\mathbf{M}\mathbf{X} - \mathbf{B}\|_2$.

*Proof.* Let $\mathbf{X}^* = \operatorname{argmin}_{\mathbf{X}} \|\mathbf{M}\mathbf{X} - \mathbf{B}\|_2$ and $\mathsf{OPT} = \|\mathbf{M}\mathbf{X}^* - \mathbf{B}\|_2$. By triangle inequality,

$$\|\mathbf{S}\mathbf{M}\mathbf{X} - \mathbf{S}\mathbf{B}\|_2 \geq \|\mathbf{S}\mathbf{M}(\mathbf{X} - \mathbf{X}^*)\|_2 - \|\mathbf{S}\mathbf{M}\mathbf{X}^* - \mathbf{S}\mathbf{B}\|_2.$$

We have

$$\frac{99}{100}\|\mathbf{M}\mathbf{X}\|_2 \leq \|\mathbf{S}\mathbf{M}\mathbf{X}\|_2 \leq \frac{101}{100}\|\mathbf{M}\mathbf{X}\|_2$$

for all $\mathbf{X} \in \mathbb{R}^{n \times m}$. Thus,

$$\|\mathbf{S}\mathbf{M}\mathbf{X} - \mathbf{S}\mathbf{B}\|_2 \geq \frac{99}{100}\|\mathbf{M}(\mathbf{X} - \mathbf{X}^*)\|_2 - \|\mathbf{S}\mathbf{M}\mathbf{X}^* - \mathbf{S}\mathbf{B}\|_2.$$

By triangle inequality,

$$\|\mathbf{S}\mathbf{M}\mathbf{X} - \mathbf{S}\mathbf{B}\|_2 \geq \frac{99}{100}\left(\|\mathbf{M}\mathbf{X} - \mathbf{B}\|_2 - \|\mathbf{M}\mathbf{X}^* - \mathbf{B}\|_2\right) - \|\mathbf{S}\mathbf{M}\mathbf{X}^* - \mathbf{S}\mathbf{B}\|_2.$$

Since $\mathbb{E}\left[\|\mathbf{S}\mathbf{v}\|_2^2\right] = \|\mathbf{v}\|_2^2$ for all $\mathbf{x} \in \mathbb{R}^d$, then by concavity and Markov's inequality, we have that

$$\mathbf{Pr}\left[\|\mathbf{S}\mathbf{M}\mathbf{X}^* - \mathbf{S}\mathbf{B}\|_2 \geq 49\|\mathbf{M}\mathbf{X}^* - \mathbf{B}\|_2\right] \leq \frac{1}{49}.$$

Thus with probability at least $0.97$,

$$\|\mathbf{S}\mathbf{M}\mathbf{X} - \mathbf{S}\mathbf{B}\|_2 \geq \frac{99}{100}\left(\|\mathbf{M}\mathbf{X} - \mathbf{B}\|_2 - \|\mathbf{M}\mathbf{X}^* - \mathbf{B}\|_2\right) - 49\|\mathbf{M}\mathbf{X}^* - \mathbf{B}\|_2.$$

Now since we have $\|\mathbf{S}\mathbf{M}\mathbf{X}^* - \mathbf{S}\mathbf{B}\|_2 \leq 49\|\mathbf{M}\mathbf{X}^* - \mathbf{B}\|_2$ and $\|\mathbf{S}\mathbf{M}\widetilde{\mathbf{X}} - \mathbf{S}\mathbf{B}\|_2 \leq \|\mathbf{S}\mathbf{M}\mathbf{X}^* - \mathbf{S}\mathbf{B}\|_2$, then

$$49\|\mathbf{M}\mathbf{X}^* - \mathbf{B}\|_2 \geq \frac{99}{100}\left(\|\mathbf{M}\widetilde{\mathbf{X}} - \mathbf{B}\|_2 - \|\mathbf{M}\mathbf{X}^* - \mathbf{B}\|_2\right) - 49\|\mathbf{M}\mathbf{X}^* - \mathbf{B}\|_2,$$

so that

$$\|\mathbf{M}\widetilde{\mathbf{X}} - \mathbf{B}\|_2 \leq 99\|\mathbf{M}\mathbf{X}^* - \mathbf{B}\|_2,$$

as desired. $\qquad \square$

**Lemma 4.3.** *Let* $\mathbf{S}, \mathbf{M}^{(1)}, \ldots, \mathbf{M}^{(\ell)}$ *be the output of Algorithm 4. Then with probability at least* $0.8$, $\max_{i \in [\ell]} \|\mathbf{A}^{(i)}\mathbf{S}\mathbf{M}^{(i)} - \mathbf{A}^{(i)}\|_F$ *is at most* $\ell^c \cdot 2^{1/c} \cdot$ $\mathcal{O}(k(\log \log k)(\log d)) \min_{\mathbf{V} \in \mathbb{R}^{k \times d}} \max_{i \in [\ell]} \|\mathbf{A}^{(i)}\mathbf{V}^\dagger \mathbf{V} - \mathbf{A}^{(i)}\|_F$.

*Proof.* Let $\widetilde{\mathbf{V}} \in \mathbb{R}^{t \times d}$ be the output of Algorithm 3, where $t = \mathcal{O}\left(k(\log\log k)(\log^2 d)\right)$. Then with probability at least $\frac{2}{3}$, we have

$$\max_{i \in [\ell]} \|\mathbf{A}^{(i)}(\widetilde{\mathbf{V}})^\dagger \widetilde{\mathbf{V}} - \mathbf{A}^{(i)}\|_F \leq \ell^c \cdot 2^{1/c} \cdot \mathcal{O}\left(k(\log\log k)(\log d)\right) \min_{\mathbf{V} \in \mathbb{R}^{k \times d}} \max_{i \in [\ell]} \|\mathbf{A}^{(i)}\mathbf{V}^\dagger \mathbf{V} - \mathbf{A}^{(i)}\|_F.$$

Therefore, we have

$$\max_{i \in [\ell]} \min_{\mathbf{B}^{(i)}} \|\mathbf{B}^{(i)}\widetilde{\mathbf{V}} - \mathbf{A}^{(i)}\|_F \leq \ell^c \cdot 2^{1/c} \cdot \mathcal{O}\left(k(\log\log k)(\log d)\right) \min_{\mathbf{V} \in \mathbb{R}^{k \times d}} \max_{i \in [\ell]} \|\mathbf{A}^{(i)}\mathbf{V}^\dagger \mathbf{V} - \mathbf{A}^{(i)}\|_F.$$

Let $p$ be a sufficiently large parameter to be fixed. By Dvoretzky's theorem, i.e., Theorem 3.2, with $\varepsilon = \mathcal{O}(1)$, we have

$$\max_{i \in [\ell]} \min_{\mathbf{B}^{(i)}} \|\mathbf{G}^{(i)}\mathbf{B}^{(i)}\widetilde{\mathbf{V}} - \mathbf{G}^{(i)}\mathbf{A}^{(i)}\|_{p,2} \leq \mathcal{O}(1) \cdot \max_{i \in [\ell]} \min_{\mathbf{B}^{(i)}} \|\mathbf{B}^{(i)}\widetilde{\mathbf{V}} - \mathbf{A}^{(i)}\|_F.$$

For sufficiently large $p = \mathcal{O}(\log \ell)$, we have by Lemma B.10,

$$\max_{i \in [\ell]} \min_{\mathbf{B}^{(i)}} \|\mathbf{B}^{(i)}\widetilde{\mathbf{V}} - \mathbf{A}^{(i)}\|_F. \leq \mathcal{O}(1) \cdot \min_{\mathbf{B}^{(i)}} \|\mathbf{G}^{(i)}\mathbf{B}^{(i)}\widetilde{\mathbf{V}} - \mathbf{G}^{(i)}\mathbf{A}^{(i)}\|_{p,2}.$$

By a change of variables, we have

$$\min_{\mathbf{X}} \|\mathbf{X}\widetilde{\mathbf{V}} - \mathbf{GA}\|_{p,2} \leq \min_{\mathbf{B}} \|\mathbf{GB}\widetilde{\mathbf{V}} - \mathbf{GA}\|_{p,2}.$$

Note that minimizing $\|\mathbf{X}\widetilde{\mathbf{V}} - \mathbf{GA}\|_{p,2}$ over all $\mathbf{X}$ corresponds to minimizing $\|\mathbf{X}_i\widetilde{\mathbf{V}} - \mathbf{G}_i\mathbf{A}\|_2$ over all $i \in [n']$. However, an arbitrary choice of $\mathbf{X}_i$ may not correspond to selecting columns of $\mathbf{A}$. Thus we apply a leverage score sampling matrix $\mathbf{S}$ to sample columns of $\widetilde{\mathbf{V}}$ which will correspondingly sample columns of $\mathbf{GA}$. Then by Lemma 4.2, we have

$$\min_{\mathbf{X}} \|\mathbf{X}\widetilde{\mathbf{V}}\mathbf{S} - \mathbf{GAS}\|_{p,2} \leq \mathcal{O}(1) \cdot \min_{\mathbf{X}} \|\mathbf{X}\widetilde{\mathbf{V}} - \mathbf{GA}\|_{p,2}.$$

Putting these together, we have

$$\min_{\mathbf{X}} \|\mathbf{X}\widetilde{\mathbf{V}}\mathbf{S} - \mathbf{GAS}\|_{p,2} \leq \ell^c \cdot 2^{1/c} \cdot \mathcal{O}\left(k(\log\log k)(\log d)\right) \min_{\mathbf{V} \in \mathbb{R}^{k \times d}} \max_{i \in [\ell]} \|\mathbf{A}^{(i)}\mathbf{V}^\dagger \mathbf{V} - \mathbf{A}^{(i)}\|_F. \quad (1)$$

Let $S$ be the selected columns from Algorithm 4 by $\mathbf{S}$ and note that $\mathbf{A}^{(i)}\mathbf{SS}^\dagger(\widetilde{\mathbf{V}})^\dagger\widetilde{\mathbf{V}}$ is in the column span of $\mathbf{A}^{(i)}\mathbf{S}$ for each $i \in [\ell]$. By Dvoretzky's theorem, i.e., Theorem 3.2, with $\varepsilon = \mathcal{O}(1)$ and a fixed parameter $p$, we have

$$\max_{i \in [\ell]} \|\mathbf{A}^{(i)}\mathbf{SS}^\dagger(\widetilde{\mathbf{V}})^\dagger\widetilde{\mathbf{V}} - \mathbf{A}^{(i)}\|_F \leq \mathcal{O}(1) \cdot \max_{i \in [\ell]} \|\mathbf{G}^{(i)}\mathbf{A}^{(i)}\mathbf{SS}^\dagger(\widetilde{\mathbf{V}})^\dagger\widetilde{\mathbf{V}} - \mathbf{G}^{(i)}\mathbf{A}^{(i)}\|_{p,2}.$$

For sufficiently large $p$, we have

$$\max_{i \in [\ell]} \|\mathbf{G}^{(i)}\mathbf{A}^{(i)}\mathbf{SS}^\dagger(\widetilde{\mathbf{V}})^\dagger\widetilde{\mathbf{V}} - \mathbf{G}^{(i)}\mathbf{A}^{(i)}\|_{p,2} \leq \mathcal{O}(1) \cdot \|\mathbf{GASS}^\dagger(\widetilde{\mathbf{V}})^\dagger\widetilde{\mathbf{V}} - \mathbf{GA}\|_{p,2}.$$

By the correctness of the leverage score sampling matrix, we have

$$\|\mathbf{GASS}^\dagger(\widetilde{\mathbf{V}})^\dagger\widetilde{\mathbf{V}} - \mathbf{GA}\|_{p,2} \leq \mathcal{O}(1) \cdot \|\mathbf{GASS}^\dagger(\widetilde{\mathbf{V}})^\dagger\widetilde{\mathbf{V}}\mathbf{S} - \mathbf{GAS}\|_{p,2}.$$

Observe that minimizing $\|\mathbf{X}\widetilde{\mathbf{V}}\mathbf{S} - \mathbf{GAS}\|_{p,2}$ over all $\mathbf{X}$ corresponds to minimizing $\|\mathbf{X}_i\widetilde{\mathbf{V}}\mathbf{S} - \mathbf{G}_i\mathbf{AS}\|_2$ over all $i \in [n']$. Moreover, the closed-form solution of the $L_2$ minization problem is

$$\mathbf{G}_i\mathbf{ASS}^\dagger(\widetilde{\mathbf{V}})^\dagger = \underset{\mathbf{X}_i}{\operatorname{argmin}} \|\mathbf{X}_i\widetilde{\mathbf{V}}\mathbf{S} - \mathbf{G}_i\mathbf{AS}\|_2.$$

Therefore, we have

$$\|\mathbf{GASS}^\dagger(\widetilde{\mathbf{V}})^\dagger\widetilde{\mathbf{V}}\mathbf{S} - \mathbf{GAS}\|_{p,2} \leq \min_{\mathbf{X}} \|\mathbf{X}\widetilde{\mathbf{V}}\mathbf{S} - \mathbf{GAS}\|_{p,2}.$$

Putting things together, we have

$$\max_{i \in [\ell]} \|\mathbf{A}^{(i)}\mathbf{SS}^\dagger(\widetilde{\mathbf{V}})^\dagger\widetilde{\mathbf{V}} - \mathbf{A}^{(i)}\|_F \leq \mathcal{O}(1) \cdot \min_{\mathbf{X}} \|\mathbf{X}\widetilde{\mathbf{V}}\mathbf{S} - \mathbf{GAS}\|_{p,2}. \quad (2)$$

By Equation 1 and Equation 2 and a rescaling of the constant hidden inside the big Oh notation, we have

$$\max_{i \in [\ell]} \|\mathbf{A}^{(i)}\mathbf{SS}^\dagger(\widetilde{\mathbf{V}})^\dagger\widetilde{\mathbf{V}} - \mathbf{A}^{(i)}\|_F \leq \ell^c \cdot 2^{1/c} \cdot \mathcal{O}\left(k(\log\log k)(\log d)\right) \min_{\mathbf{V} \in \mathbb{R}^{k \times d}} \max_{i \in [\ell]} \|\mathbf{A}^{(i)}\mathbf{V}^\dagger\mathbf{V} - \mathbf{A}^{(i)}\|_F.$$

The desired claim then follows from the setting of $\mathbf{M}^{(i)} = \mathbf{S}^\dagger(\widetilde{\mathbf{V}})^\dagger\widetilde{\mathbf{V}}$ for $i \in [\ell]$ by Algorithm 4. $\quad\square$

# D   Additional Empirical Evaluations

In this section, we present a number of additional results from our empirical evaluations on socially fair low-rank approximation.

We first remark on a number of additional empirical evaluations for socially fair low-rank approximation. We first compare our bicriteria algorithm to the standard low-rank approximation baseline over a simple synthetic example. We then describe a simple fixed input matrix for socially fair low-rank approximation that serves as a proof-of-concept similarly demonstrating the improvement of fair low-rank approximation optimal solutions over the optimal solutions for standard low-rank approximation.

**Synthetic dataset.**   Next, we show that for a simple dataset with two groups, each with two observations across four features, the performance of the fair low-rank approximation algorithm can be much better than standard low-rank approximation algorithm on the socially fair low-rank objective, even without allowing for bicriteria rank. For our synthetic dataset, we generate simple $\mathbf{A}^{(1)} = \begin{bmatrix} 2 & 0 & 0 & 0 \\ 0 & 2 & 0 & 0 \end{bmatrix}$ and $\mathbf{A}^{(2)} = \begin{bmatrix} 0 & 0 & 1.99 & 0 \\ 0 & 0 & 0 & 1.99 \end{bmatrix}$. An optimal fair low-rank solution for $k = 2$ is $\begin{bmatrix} 1 & 0 & 0 & 0 \\ 0 & 0 & 1 & 0 \end{bmatrix}$, which gives cost 4. However, due to the weight of the items in $\mathbf{A}^{(1)}$, the standard low-rank algorithm will output low-rank factors with cost $2 \cdot 1.99^2 \approx 7.92$, such as the factors $\begin{bmatrix} 1 & 0 & 0 & 0 \\ 0 & 1 & 0 & 0 \end{bmatrix}$. Thus the ratio between the objectives is roughly $\frac{1}{2}$, i.e., the improvement is roughly a factor of two. We show in Figure 2 that our bicriteria algorithm achieves similar improvement.

**Experimental setup.**   Again, we compare our bicriteria algorithm from Algorithm 3 against the standard non-fair low-rank approximation algorithm that outputs the top $k$ right singular vectors from the singular value decomposition. Per Dvoretzky's Theorem, c.f., Theorem 3.2, we generate normalized Gaussian matrices $\mathbf{G}$ and $\mathbf{H}$ of varying size and then use $L_p$ Lewis weight sampling to generate a matrix $\mathbf{T}$ with varying values of $p$. We generate matrices $\mathbf{T}$, $\mathbf{G}$, and $\mathbf{H}$ with a small number of dimensions and thus do not compute the sampling matrix $\mathbf{S}$ but instead use the full matrix. We first fix $p = 1$ and iterate over the number of rows/columns in the Gaussian sketch in the range $\{1, 2, \ldots, 19, 20\}$ in Figure 2a. We then fix the Gaussian sketch to have three rows/columns and iterate over $p \in \{1, 2, \ldots, 9, 10\}$ in Figure 2b. For each of the variables, we compare the outputs of the two algorithms across 100 iterations and plot the ratio of their fair costs. In particular, we set the same rank parameter of $k = 2$ for both algorithms; we remark that the theoretical guarantees for our bicriteria algorithm are even stronger when we permit the solution to have rank $k'$ for $k' > k$.

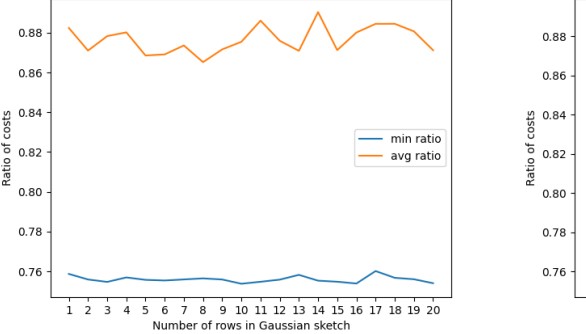
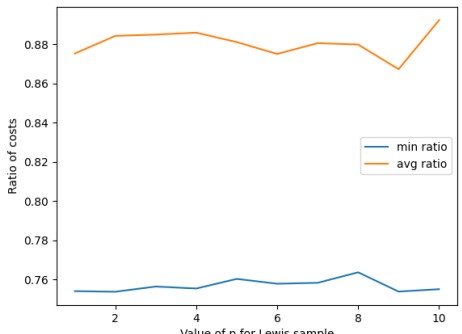

(a) Ratio of costs over different number of rows/columns in the Gaussian sketch matrix.

(b) Ratio of costs over different values of $p$ in the Lewis sampling process.

Fig. 2: Ratio of the cost of our bicriteria algorithm to the cost of the standard low-rank approximation solution for $k = 2$, across 100 iterations.

**Results and discussion.** Our empirical evaluations in Figure 2 show that our algorithms can perform significantly better for socially fair low-rank approximation. We note that in both across all values of the number of rows/columns in the Gaussian sketch in Figure 2a and across all values of $p$ in the Lewis weight sampling parameter in Figure 2b, the average ratio between our bicriteria solution and the standard low-rank approximation is less than 0.9. We remark that any ratio less than 1 demonstrates the superiority of our algorithm, with smaller ratios indicating better performance. In fact, the minimum ratio is as low as 0.76.

**Proof-of-concept.** Finally, we give a toy example using a synthetic dataset showing the importance of considering fairness in low-rank approximation. Namely, we show that even for a simple dataset with four groups, each with a single observation across two features, the performance of the fair low-rank approximation algorithm can be much better than standard low-rank approximation algorithm on the socially fair low-rank objective. In this setup, we repeatedly generate matrices $\mathbf{A}^{(1)}, \mathbf{A}^{(2)}, \mathbf{A}^{(3)}, \mathbf{A}^{(4)} \in \{0,1\}^2$, with $\mathbf{A}^{(1)} = (1,0)$ and $\mathbf{A}^{(2)} = \mathbf{A}^{(3)} = \mathbf{A}^{(4)} = (0,1)$. The optimal fair low-rank solution is $\left(\frac{\sqrt{2}}{2}, \frac{\sqrt{2}}{2}\right)$ but due to the extra instances of $(0,1)$, the standard low-rank algorithm will output the factor $(0,1)$. Thus the optimal fair solution achieves value $\frac{1}{4}$ on the socially fair low-rank approximation objective while the standard low-rank approximation solution achieves value $\frac{1}{2}$, so that the ratio is a $50\%$ improvement.

## Potential Broader Impact

This paper presents research aimed at reducing biases in key data summarization tasks. Our work is driven by the societal implications of machine learning techniques, representing a series of results towards developing more equitable methods in this field.

