# OpenReview forum: "On Socially Fair Low-Rank Approximation and Column Subset Selection"
_NeurIPS.cc/2024/Conference — NeurIPS 2024 poster_

### Official Review · Reviewer_aAWQ · 2024-07-11

**Soundness:** 3
**Presentation:** 3
**Contribution:** 3
**Rating:** 7
**Confidence:** 4

**Summary:**

This paper considers the fair low-rank approximation and fair column subset selection problem. These two problems are similar; they aim to select a subset of vectors that optimize the algorithm's performance across all sub-populations. Formally, given $\ell$ matrices A and $\ell$ matrices B, the goal is to choose k vectors that span these $\ell$ B matrices such that the maximum distance between A and B is minimized. The distance is defined as some norm function between these $2\ell$ matrices. Such a problem has applications in feature selection, which is one of the most important topics in machine learning.

The main contributions of this work are (1) a $(1+\epsilon)$-approximation algorithm running in O(2^n); (2) using the similar key idea, one can obtain an $\tilde{O}(k)$-approximation running in polynomial time for column selection. The authors also show a lower bound that achieving constant approximation requires exponential running time under the ETH conjecture. Besides these theoretical results, the authors also provide a set of experimental results.

**Strengths:**

1. The studied problem is well-motivated and it has wide application in machine learning. I believe that it should be interesting in the ML community.

2. This is a technical solid paper. I like the high-level idea of the algorithm. Namely, it starts from some "bad" solution, and then repeatedly decreases the approximation until the ratio becomes $(1+\epsilon)$. Checking feasibility requires some new ideas.

3. The paper is well-written. I appreciate that the authors provide sufficient intuition and high-level descriptions for algorithms in Section 1.1. They are very helpful.

**Weaknesses:**

The main weakness is that the $(1+\epsilon)$-approximation runs in exponential time, while for the column selection algorithm, its running time is polynomial but the approximation ratio is linear in k. I understand that there is a lower bound under ETH conjecture, but this running time strictly restricts the applications to the algorithm. Maybe it is more suitable to study the parallel algorithms for these problems.

**Questions:**

Is it possible to make the proposed algorithms run in parallel?

**Limitations:**

This is a theoretical paper, there are no potential negative societal impacts.

---

> ### Author Rebuttal · Authors · 2024-08-06
>
> > The main weakness is that the $(1+\epsilon)$-approximation runs in exponential time while for the column selection algorithm, its running time is polynomial but the approximation ratio is linear in $k$. I understand that there is a lower bound under ETH conjecture, but this running time strictly restricts the applications to the algorithm. Maybe it is more suitable to study the parallel algorithms for these problems... Is it possible to make the proposed algorithms run in parallel?
>
> Thanks for the suggestion. It could be possible to make the proposed algorithms run in parallel, but due to our hardness results, the total runtime for a $(1+\epsilon)$-approximation algorithm must still be exponential, across all servers. That said, this is a very interesting future direction.
>
> Additionally, in most practical instances, $k$ is usually a small constant. Therefore, for these instances our algorithm provides a polynomial time constant factor approximations.

---

> > ### Comment · Reviewer_aAWQ · 2024-08-11
> >
> > I'd like to thank the authors for replying to my concerns. I will keep my score unchanged. Please add some descriptions for the scenarios where k is a small constant.

---

### Official Review · Reviewer_CrGT · 2024-07-11

**Soundness:** 3
**Presentation:** 2
**Contribution:** 3
**Rating:** 7
**Confidence:** 3

**Summary:**

The paper considers low-rank approximation and column selection in a specific setting to which authors refer as socially fair setting. Basically, given $\ell$ matrices $A^{(1)},\dots,A^{(\ell)}$, they consider the problem of approximating solution of $\\min_{U \\in \\mathbb{R}^{k\\times d}} \\max_{i\\in [\ell]} \\Vert A^{(i)}U^{\\dagger}U - A^{(i)}\\Vert_F$. They first show that this problem is in general NP-hard, and then propose two algorithms that achieve time complexity polynomial in $n=\sum_{i=1}^{\ell} n_i$ and either exponential in $k$ (rank) for $(1+\epsilon)$ approximation, or polynomial in $k$ for an approximation with multiplicative constant polynomial in $k,\ell,\log d$. Lastly, proposed algorithm for low rank approximation is then used for column subset selection, where similar guarantees are obtained.

**Strengths:**

1) This paper has a few interesting and nontrivial contributions. The way these are presented is also very good - starting from computationally infeasible (in polynomial time) results, to less time demanding algorithms.

2) I believe the authors made a nice connection with fairness, and this clearly increases significance of the paper. Nonetheless, I believe this is a fundamental problem of multi-matrix approximation and I appreciate derived results even from pure theoretical perspective.

3) It seems to be the first paper considering socially fair low-rank matrix approximation and related topics.

4) Although the main results and their analysis are not simple, I believe the authors made great efforts to make it as accessible as possible.

**Weaknesses:**

1) There are a lot of things to understand in Introduction i.e. section 1.1. Although it gets easier once you are done with it, I would prefer keeping some of the details of section 1.1 until section 3. Furthermore, there is a significant overlap between the two sections, so merging them might give you more space for explanations in the main.

**Questions:**

1) According to Theorem 1.3 the runtime is given by $\\frac{1}{\epsilon} \mathrm{poly}(n) (2\ell)^{O(N)}$ where $N=\mathrm{poly}(\ell,k,\frac{1}{\epsilon})$. You present this result for fixed number of groups $\ell$ and as a function of $n$ and $k$. But in the case when all matrices are (approximately) low rank (say of rank $r$) then it does not make sense to have $k\geq r\ell$? So, in this case we are equally interested in dependence on $\ell$, right?

2)  Notation in line (286) is confusing - I am not sure if $B^{(i)}$ is defined at that point, and how $B^{(i)} C$ becomes $A^{(i)}SM^{(i)}$ in Lemma 4.3. In addition, it seems like $M^{(1)}=\cdots=M^{(\ell)} = M$, so why are there these superscripts?

3) In the setting when $\ell = 1$, your column selection algorithm has the same (or higher) time complexity as SVD, but has multiplicative factor $\widetilde{O}(k)$ in front of the approximation error. Thus, in this setting your algorithm is as complex as the optimal algorithm, but achieves worse performance. But, in any setting $\ell > 1$, it is not clear how to combine subspaces obtained from SVD to achieve small fair low-rank approximation error. Is this correct?

4) Could you please explain how you choose rank for bicriteria approximation in Section 5? Namely, at the end of Section 1.1 you say that bicriteria algorithm performs better "even when the bicriteria algorithm is not allowed a larger rank than the baseline". Later in Section 5 you choose bicriteria solution to have rank $2k$. I find this a bit confusing, especially since I thought $k$ was the rank...


Other comments
- missing definition of $k$ in the abstract
- there is a typo in Lemma 3.5, on the right hand side of the last inequality it should be $\\min_V$ and not $\tilde{V}$.
- in Section 5 I did not find definition of the costs ratio you have plotted (I believe it is given in Appendix), so please add it there.

**Limitations:**

The authors have addressed limitations adequately.

---

> ### Author Rebuttal · Authors · 2024-08-06
>
> > There are a lot of things to understand in Introduction i.e. section 1.1. Although it gets easier once you are done with it, I would prefer keeping some of the details of section 1.1 until section 3. Furthermore, there is a significant overlap between the two sections, so merging them might give you more space for explanations in the main.
>
> We will apply your suggestion in the next version of our paper, thank you.
>
> > According to Theorem 1.3 the runtime is given by $\frac{1}{\epsilon}\text{poly}(n)(2\ell)^{O(N)}$ where $N=\text{poly}\left(\ell,k,\frac{1}{\epsilon}\right)$. You present this result for fixed number of groups $\ell$ and as a function of $n$ and $k$. But...in the case when all matrices are (approximately) low rank (say of rank $r$) then it does not make sense to have $k\ge r\ell$?...So, in this case we are equally interested in dependence on $\ell$, right?
>
> Even in this case, we believe that the focus can be on the input parameter $k$. In particular, in the case where all matrices have rank at most $r$, the input value of $k$ should not be set larger than $r\ell$. Indeed, given a more reasonable input parameter of $k<r\ell$ that does not trivialize the problem, the algorithmic runtime dependency on $k$ is still quite important.
>
> > Notation in line (286) is confusing - I am not sure if $B^{(i)}$ is defined at that point, and how $B^{(i)}C$ becomes $A^{(i)}SM^{(i)}$ in Lemma 4.3. In addition, it seems like $M^{(1)}=\cdots=M^{(\ell)}=M$, so why are there these superscripts?
>
> Thanks for the question, there is a slight typo -- the expression in Line 286 should be $||A^{(i)}CB^{(i)}-A^{(i)}||_F$. We are selecting $k$ columns of the overall matrix $A$, which also corresponds to the same $k$ columns for each group $A^{(i)}$. Note that $B^{(i)}$ is defined in this optimization problem as the minimizing factor. Given the selection of the $k$ columns induced by $C$, then $B^{(i)}$ is the matrix that minimizes the low-rank cost for $A^{(i)}$. We then use a change of notation, the column selection matrix $C$ becomes the column sampling matrix $S$ and $B^{(i)}$ becomes $M^{(i)}$, so that the matrices $M^{(1)},\ldots,M^{(\ell)}$ differ across the groups. We will fix the typo and unify the notation in the next version of the paper.
>
> > In the setting when $\ell=1$, your column selection algorithm has the same (or higher) time complexity as SVD, but has multiplicative factor $\tilde{O}(k)$ in front of the approximation error...Thus, in this setting your algorithm is as complex as the optimal algorithm, but achieves worse performance. But, in any setting $\ell>1$, it is not clear how to combine subspaces obtained from SVD to achieve small fair low-rank approximation error. Is this correct?
>
> Note that even for $\ell=1$, SVD does not translate to column subset selection since the top $k$ singular values generally do not correspond to elementary vectors (i.e., columns of the matrix), though SVD does give the optimal solution for low-rank approximation. More generally, it is not even clear how to combine subspaces acquired from SVD for the socially fair low-rank approximation with $\ell>1$.
>
> > Could you please explain how you choose rank for bicriteria approximation in Section 5?...Namely, at the end of Section 1.1 you say that bicriteria algorithm performs better ``even when the bicriteria algorithm is not allowed a larger rank than the baseline''. Later in Section 5 you choose bicriteria solution to have rank $2k$. I find this a bit confusing, especially since I thought $k$ was the rank...
>
> We perform multiple experiments in Section 5. In Figure 1a, the bicriteria algorithm is not allowed a larger rank than the baseline. In Figure 1b, the bicriteria algorithm is permitted rank $2k$, where $k$ is the rank of the baseline.

---

> ### Comment · Reviewer_CrGT · 2024-08-12
>
> Thank you for your reply. I believe this is a strong paper and I will maintain my score.

---

### Official Review · Reviewer_YsQa · 2024-07-12

**Soundness:** 2
**Presentation:** 2
**Contribution:** 2
**Rating:** 4
**Confidence:** 3

**Summary:**

The authors investigate socially-fair low-rank approximation and column subset selection problems. The concept of socially-fair (in the context of clustering problems this fairness notion is well studied) is introduced when the input matrix rows can be partitioned resulting in sub-matrices, the goal is to find a low-rank matrix of rank $k$ that minimizes the maximum reconstruction error or loss. In the socially-fair column subset selection problem, the task is to select $k$ columns from the input matrix that minimize the maximum reconstruction error. The quality of the solution (error) is measured by the residual norm when the input matrix is projected onto the subspace spanned by the selected columns.

First they show that fair low-rank approximation is NP-hard to approximate to any constant factor, further they strengthen their inapproximability claim by showing that it is not possible to find an algorithm with running time $2^{\Omega(k)}$ that gives a constant factor approximation. Note that these results only apply for low-rank approximation and not column-subset selection (correct me if I am wrong)---these proofs/construction was easier to follow and I have verified in sufficient detail, they appear to be correct. They proceed to present approximation algorithms, asserting that these algorithms run in time $2^{\text{poly}(k)}$. I have not been able to verify these proofs myself.

**Strengths:**

The paper presents approximation algorithms for the socially-fair low-rank approximation and column subset selection problem, which are both important from the context of dimensionality reduction and algorithmic fairness perspective.

**Weaknesses:**

The paper seems to be written for a specialized audience deeply embedded in the field, rather than for a general audience. The authors frequently cite lemmas and theorems from previous papers without providing sufficient explanations or context, assuming readers already have extensive background knowledge. This approach can alienate those who are not experts in the niche area of matrix approximations. The writing style is convoluted, making it challenging to follow the arguments and understand the implicit assumptions. The authors should consider that not everyone working on these topics is an expert with a long history in the field, and make an effort to present the material more clearly and accessibly.

I was not able to understand the paper in sufficient detail to provide the best possible review. I cannot confirm that the proofs are correct, as I could not verify them without clarifying several unclear aspects of the paper. However, if the authors can address my questions in sufficient detail, I am willing to review the paper again and update my assessment.

**Questions:**

Line 58: $U = U_1 \circ \dots \circ U_k$, what operations does $\circ$ symbol denote?

Line 82: Why does the naive algorithm require $ n^{\text{poly}(k)} $ and not $ n^k \text{poly}(n, k) $? Can you clarify this precisely? Which specific naive algorithm are you referring to here?

Line 88-92: I'm having trouble understanding this. Could you explain what this value of $ \alpha $ is and how it relates to Theorem 1.3 in simpler terms? Once this is clear, the subsequent statements will be easier to follow. The theorem statement does not mention the term $ \alpha $; how does it connect to $ \alpha $? What do you mean when you say $ \alpha $ is feasible?

Line 131: Could you please provide a precise definition of the fair column subset selection problem, or indicate where it is defined in the paper, before discussing algorithmic results for the problem?

Line 183: Is the problem studied by Matakos et al. 2023 the same or "similar" to the problem addressed in this paper? What are the specific differences between them? Specifically, does this paper generalize the problem defined on two groups by Matakos et al. to more than two groups, or is it a different notion of fairness altogether?

Line 206 - 228: It is redundant to repeat the exact same text as outlined in our contributions and technical overview. I am not sure if this is the most efficient way to utilize the space by reiterating these statements multiple times, there is nothing new, the explanation is again in high-level without any details.

Line 286: What is the relationship between matrix $ B^{(i)} $ and $ A^{(i)} $? Where is $ B^{(i)} $ defined in the paper? Without clarifying this, I cannot proceed to verify the subsequent details. Are you selecting $ C $ as a column matrix of $ A $ or $ A^{(i)} $? I am concerned that the dimensions of matrices may not match with the current problem formulation.

**Limitations:**

No discussion on potential negative social impact.

---

> ### Author Rebuttal · Authors · 2024-08-06
>
> > The paper seems to be written for a specialized audience deeply embedded in the field, rather than for a general audience. The authors frequently cite lemmas and theorems from previous papers without providing sufficient explanations or context, assuming readers already have extensive background knowledge. This approach can alienate those who are not experts in the niche area of matrix approximations. The writing style is convoluted, making it challenging to follow the arguments and understand the implicit assumptions. The authors should consider that not everyone working on these topics is an expert with a long history in the field, and make an effort to present the material more clearly and accessibly.
>
> We emphasize that to provide context, the previous version already includes simple intuitive explanations prior to each lemma and theorem statement, even those from previous papers. Moreover, the nature of our result is theoretical and, in principle, builds on technical results in the area of randomized numerical linear algebra. However, we will provide more background and preliminaries in the next version of our paper.
>
> > Line 58: $U=U_1\circ\cdots\circ U_k$, what operations does $\circ$ symbol denote?
>
> $\circ$ denotes vertical concatenation of the rows, c.f., line 542 in the preliminaries.
>
> > Line 82: Why does the naive algorithm require $n^{\mathrm{poly}(k)}$ and not $n^k\mathrm{poly}(n,k)$? Can you clarify this precisely? Which specific naive algorithm are you referring to here?
>
> The algorithm we were referring to is a brute force search over a net on all subsets of $k$ factors, which for constant dimension would require $n^{\mathrm{poly}(k)}$ time (and even worse for super-constant dimension). Moreover, it should be noted that a runtime of $n^k\mathrm{poly}(n,k)$ would still fall under $n^{\mathrm{poly}(k)}$ runtime, i.e., the polynomial would be linear.
>
> > Line 88-92: I'm having trouble understanding this. Could you explain what this value of $\alpha$ is and how it relates to Theorem 1.3 in simpler terms? Once this is clear, the subsequent statements will be easier to follow. The theorem statement does not mention the term $\alpha$; how does it connect to $\alpha$? What do you mean when you say $\alpha$ is feasible?
>
> $\alpha$ is simply a guess for a $(1+\epsilon)$-approximation to the optimal solution. The theorem does not need to mention the term $\alpha$ because the algorithm makes a small number of guesses, one of which must be feasible, i.e., a $(1+\epsilon)$-approximation to the optimal solution for which there exists $k$ factors that realize the fair low-rank approximation cost.
>
> > Line 183: Is the problem studied by Matakos et al. 2023 the same or ``similar'' to the problem addressed in this paper? ... What are the specific differences between them? Specifically, does this paper generalize the problem defined on two groups by Matakos et al. to more than two groups, or is it a different notion of fairness altogether?
>
> Matakos et al. (2023) study fair column subset selection, which is a specific type of low-rank approximation. Thus, their focus is more narrow than ours. They present an algorithm that can only achieve fairness for two groups, which may be inapplicable for many applications. Therefore, it is accurate to say that our setting for the specific problem of socially fair column subset selection generalizes the problem they study, although the techniques in the corresponding algorithms are quite different. To emphasize, we both study the same notion of fairness.
>
> > Line 206 - 228: It is redundant to repeat the exact same text as outlined in our contributions and technical overview...I am not sure if this is the most efficient way to utilize the space by reiterating these statements multiple times, there is nothing new, the explanation is again in high-level without any details.
>
> This section expands on the algorithmic outline given in the technical overview to provide more intuition. While we understand that some details may appear repetitive, this is necessary to maintain coherence and clarity for the reader, especially when transitioning between different sections of the paper. That said, we will use the extra page in the introduction to address your comments. Additionally, we provide the full details in Algorithms 1 and 2 on Line 238.
>
> > Line 286: What is the relationship between matrix $B^{(i)}$ and $A^{(i)}$?...Where is $B^{(i)}$ defined in the paper? Without clarifying this, I cannot proceed to verify the subsequent details. Are you selecting $C$ as a column matrix of $A$ or $A^{(i)}$? I am concerned that the dimensions of matrices may not match with the current problem formulation.
>
> Ah thanks, there is a slight typo --  the expression in Line 286 should be $\|A^{(i)}CB^{(i)}-A^{(i)}\|_F$ and hence the dimensions of the matrices match the current problem formulation. We are selecting $k$ columns of the overall matrix $A$, which also corresponds to the same $k$ columns for each group $A^{(i)}$. Note that $B^{(i)}$ is defined in this optimization problem as the minimizing factor. Given the selection of the $k$ columns induced by $C$, then $B^{(i)}$ is the matrix that minimizes the low-rank cost for $A^{(i)}$.

---

> > ### Comment · Reviewer_YsQa · 2024-08-08
> > **Response to rebuttal**
> >
> > **["slight typo" in Line 286]** In the current formulation of the submitted paper, the dimensions of the matrices do not match, which the authors have casually dismissed this as a "slight typo." This, however, is a significant error, raising concerns about whether the proofs in Section 4 were thoroughly read and verified. Given that the objective being minimized was not accurately stated, it is unclear how one could have validated the claimed approximation ratios and algorithmic results in that section.
> >
> > **[improving writing]** The authors do not seem fully committed to revising the writing to enhance the accessibility of their research, as the descriptions currently lack clarity. In my view, the paper requires significant revision to be accessible to a wider audience. Unless the authors address these concerns and demonstrate a strong commitment to improving the writing, I would maintain my current evaluation.

---

> > > ### Author Response · Authors · 2024-08-08
> > >
> > > Thank you for the follow-up questions! We hope the following clarifications address your points; if not, we would also be happy to continue discussing any possible concerns.
> > >
> > > > ["slight typo" in Line 286] In the current formulation of the submitted paper, the dimensions of the matrices do not match, which the authors have casually dismissed this as a "slight typo." This, however, is a significant error, raising concerns about whether the proofs in Section 4 were thoroughly read and verified. Given that the objective being minimized was not accurately stated, it is unclear how one could have validated the claimed approximation ratios and algorithmic results in that section.
> > >
> > > We would like to emphasize that the objective function precisely matches both the informal description of the problem, i.e., "to select $k$ columns", and the subsequent analysis. In particular, the cost $||A^{(i)}CB^{(i)}-A^{(i)}||_F$ precisely matches the cost in Lemma 4.3 $||A^{(i)}SM^{(i)}-A^{(i)}||_F$ up to a change of notation (where $C$ is $S$ and $B^{(i)}$ is $M^{(i)}$). Note that $C\in\mathbb{R}^{d\times k}$ and $B^{(i)}\in\mathbb{R}^{k\times d}$ for all $i\in[\ell]$, so that $A^{(i)}CB^{(i)}$ has dimension $n_i\times d$, which is the same as the dimension of $A^{(i)}$. Therefore, the dimensions match.
> > >
> > > For the sake of consistency, we further remark that Reviewer CrGT raised a similar question about the objective and we provided a response that exactly matches these details, i.e., see the response to Reviewer CrGT (https://openreview.net/forum?id=EO1Qev952p&noteId=A1p5yN7fG0)
> > >
> > > > [improving writing] The authors do not seem fully committed to revising the writing to enhance the accessibility of their research, as the descriptions currently lack clarity. In my view, the paper requires significant revision to be accessible to a wider audience. Unless the authors address these concerns and demonstrate a strong commitment to improving the writing, I would maintain my current evaluation.
> > >
> > > We apologize that our initial response conveys this sentiment. We are fully committed to producing a manuscript that is accessible to the general research community and in fact, we have already incorporated the explicit suggestions from the initial reviews. We are also taking additional more thorough passes to improve the overall exposition of the paper, as well as preparing a "full version" with the complete details and additional intuition in the main body. We hope these updates will address the remaining concerns.

---

> ### Comment · Reviewer_YsQa · 2024-08-12
> **Re: followup for official comment(s)**
>
> I thank authors for their responses.
>
> After carefully checking the rebuttal and considering the comments from other reviewers, I went through the paper again. Unfortunately, I still find it challenging to follow the arguments and writing due to inconsistent and confusing notation. For example, the matrix $A$ is sometimes referred to as $M$, with corresponding changes from $A^{(i)}$ to $M^{(i)}$, and the column matrix is inconsistently labeled as $C$, $V$, or $S$ across different sections. $B^{(i)}$ is used before it is defined. In Lemma 3.5, $c$ is not defined; one has to go to the proof of the Lemma in Line 725 to even known what is $c$ or refer back to the statement in Theorem 1.4. Such inconsistency makes the paper difficult to read and understand. Also, the equations are poorly organized and spread across multiple lines. I often had to rewrite these equations on a paper to make sense of them.
>
> I acknowledge that the paper may have strong theoretical contributions, and if all the claimed results are correct, they are significant. However, despite my experience with similar topics, I struggled to follow the proof arguments due to unclear writing and constant change in notation. The authors should recognize that this level of writing is not ideal and likely falls short of the standard while reviewing. Sections 3, 4, and 5 are particularly problematic. Even after multiple clarifications, these sections remain difficult to read and understand. While the statements and approaches appear correct at a high level, verifying the proofs becomes tedious due to the way the it is presented, making it difficult to check the details precisely.
>
> Being said that, I am not fully confident about the correctness of the proofs, so I will retain my original evaluation. To be clear, my evaluation is based on the difficulty in validating the claims due to the imprecise writing. If the theoretical contributions are correct, this work is a valuable addition in the study of fair variants of matrix approximation problems. I hope other reviewers have been able to thoroughly check the proofs in ways I could not.
>
> Regarding the proof arguments in Section 5, they seem to hold at a high level. Also, Matakos et al. (2023) have established a lower bound on the number of columns, $k \ell$, and choosing fewer columns than this would make it unbounded. Therefore, the results for column subset selection are only meaningful if $\ell \gg \log k$ (a similar comment also noted by reviewer CrGT in the context of Theorem~1.3).

---

> > ### Author Response · Authors · 2024-08-13
> >
> > Thank you for both the continued correspondence and the time for taking an additional pass over our paper. We really appreciate the extra effort by the referee toward the overall review process -- our paper will certainly benefit from the thorough feedback.
> >
> > >...For example, the matrix $A$ is sometimes referred to as $M$, with corresponding changes from $A^{(i)}$ to $M^{(i)}$ and the column matrix is inconsistently labeled as $C$, $V$, or $S$ across different sections. $B^{(i)}$ is used before it is defined. In Lemma 3.5, $c$ is not defined; one has to go to the proof of the Lemma in Line 725 to even known what is $c$ or refer back to the statement in Theorem 1.4.
> >
> > For what it's worth, our intention is to use $A$ as the input matrix to the problem and $M$ as an input matrix for leverage score sampling, the matrices $\{A^{(i)}\}$ as the input groups, and the matrices $\{M^{(i)}\}$ as outputs to Algorithm 4. We will unify $C$ and $S$ as the column selection matrix, and our intention is to use $V$ as a set of rank-$k$ factors that need not be restricted to a subset of columns. We will make the implicit definition of $B^{(i)}$ in Line 286 more explicit. Finally, we agree that the trade-off parameter $c<1$ is not clearly stated prior to Lemma 3.5. We apologize for the confusion -- we will clarify the purpose of $c$ in the statement of Lemma 3.5, as well as the surrounding context.
> >
> > > Also, Matakos et al. (2023) have established a lower bound on the number of columns, $k\ell$, and choosing fewer columns than this would make it unbounded. Therefore, the results for column subset selection are only meaningful if $\ell\gg\log k$ (a similar comment also noted by reviewer CrGT in the context of Theorem~1.3).
> >
> > Actually, while Matakos et al. (2023) considers the same intuitive goal of choosing $k$ columns to minimize the maximum cost, their cost function is normalized by $\frac{1}{||A^{(i)}-A^{(i)}_k||_F}$, where $A^{(i)}_k$ is the best rank-$k$ approximation to $A^{(i)}$. Therefore, the lower bound of Matakos et al. (2023) does not apply to our objective.
> >
> > For example, it can be shown that returning the best $k$ columns for our column subset selection objective on the entire matrix $A=A^{(1)}\circ\ldots\circ A^{(\ell)}$ is an $\ell$-approximation to the fair column subset selection problem. Thus, it is not necessary to choose $k\ell$ columns and more generally, our results for column subset selection do actually hold across all settings of $\ell$.
> >
> > > I acknowledge that the paper may have strong theoretical contributions, and if all the claimed results are correct, they are significant. However, despite my experience with similar topics, I struggled to follow the proof arguments due to unclear writing and constant change in notation. The authors should recognize that this level of writing is not ideal and likely falls short of the standard while reviewing. Sections 3, 4, and 5 are particularly problematic. Even after multiple clarifications, these sections remain difficult to read and understand. While the statements and approaches appear correct at a high level, verifying the proofs becomes tedious due to the way the it is presented, making it difficult to check the details precisely.
> >
> > Thank you for your positive assessment of the theoretical contributions. We emphasize that we are fully committed to improving the overall presentation of the paper, so that it is accessible to the general research community.

---

> > > ### Comment · Reviewer_YsQa · 2024-08-13
> > > **Re: Official comment**
> > >
> > > **[Lower bound on number of columns: Actually, while Matakos et al. (2023) considers $\dots$ ]** I agree with the authors on this point, and they are correct.

---

### Official Review · Reviewer_icAD · 2024-07-15

**Soundness:** 3
**Presentation:** 3
**Contribution:** 3
**Rating:** 7
**Confidence:** 2

**Summary:**

The paper studies the socially-fair variants of low-rank approximation and column subset selection. The authors prove hardness results similar to those in the literature for the non-fair versions of the problems. They then propose exponential time close-to-optimal solution for socially-fair low rank approximation and polynomial time bi-criteria approximation algorithms for both the problems.

**Strengths:**

1. The paper studies two important and relevant problems.
2. The socially-fair variants are established notions of fairness in clustering, dimension reduction etc., and applies very well to both low rank approximation and column subset selection, neither of which are solved in the literature.
3. The theoretical analysis is well done.
4. The experimental results show that the algorithms are practical enough to use in the real-world applications.

**Weaknesses:**

The paper is not easy to read. The introduction may be shortened.

**Questions:**

Are there any open problems related to socially-fair low rank approximation and subset selection not addressed in the paper? If yes, I request the authors to mention some.

**Limitations:**

Limitations adequately addressed by the authors.

---

> ### Author Rebuttal · Authors · 2024-08-06
>
> > The paper is not easy to read. The introduction may be shortened.
>
> The nature of our result is theoretical and, in principle, builds on technical results in the area of randomized numerical linear algebra. However, we will provide more background and preliminaries in the next version of our paper.
>
> > Are there any open problems related to socially-fair low rank approximation and subset selection not addressed in the paper? If yes, I request the authors to mention some.
>
> Thank you for your suggestions. We will add related open problems in the next version of our paper, including both technical questions and more general fairness-aware optimization of data summarization methods. For example, a natural future direction is the efficient construction of $(1+\epsilon)$-coresets with minimal size for socially fair subset selection.

---

### Author Rebuttal · Authors · 2024-08-06

We thank the reviewers for their thoughtful comments and valuable feedback. We also appreciate the positive remarks, such as:
- The paper studies two important and relevant problems. (Reviewer icAD)
- The theoretical analysis is well done. (Reviewer icAD)
- The experimental results show that the algorithms are practical enough to use in the real-world applications. (Reviewer icAD)
- The paper presents approximation algorithms for the socially-fair low-rank approximation and column subset selection problem, which are both important from the context of dimensionality reduction and algorithmic fairness perspective. (Reviewer YsQa)
- This paper has a few interesting and nontrivial contributions. The way these are presented is also very good - starting from computationally infeasible (in polynomial time) results, to less time demanding algorithms. (Reviewer CrGT)
- I believe the authors made a nice connection with fairness, and this clearly increases significance of the paper. Nonetheless, I believe this is a fundamental problem of multi-matrix approximation and I appreciate derived results even from pure theoretical perspective. (Reviewer CrGT)
- It seems to be the first paper considering socially fair low-rank matrix approximation and related topics. (Reviewer CrGT)
- Although the main results and their analysis are not simple, I believe the authors made great efforts to make it as accessible as possible. (Reviewer CrGT)
- The studied problem is well-motivated and it has wide application in machine learning. I believe that it should be interesting in the ML community. (Reviewer aAWQ)
- This is a technical solid paper. I like the high-level idea of the algorithm. Namely, it starts from some "bad" solution, and then repeatedly decreases the approximation until the ratio becomes $(1+\epsilon)$. Checking feasibility requires some new ideas. (Reviewer aAWQ)
- The paper is well-written. I appreciate that the authors provide sufficient intuition and high-level descriptions for algorithms in Section 1.1. They are very helpful. (Reviewer aAWQ)

We provide our responses to the initial comments of each reviewer separately below. We hope our answers addresses all points raised by the reviewers and we will be most happy to answer any remaining or additional questions during the discussion phase!

---

### Decision · Program_Chairs · 2024-09-25

**Decision:**

Accept (poster)

**Comment:**

The reviewers agree that this paper studies important problems that lie in the intersection of algorithmic fairness and low-rank approximation. They acknowledged that the paper is technically solid and novel, and also considered the exposition of the techniques to be reasonably good. Experimental results serve as a good addition to the theory. Although one reviewer still has concerns on writing consistencies in the discussion stage, overall the pros significantly outweigh the cons.